# Three dimensional reconstruction of energy stores for jumping in planthoppers and froghoppers from confocal laser scanning microscopy

Igor Siwanowicz[1]\*, Malcolm Burrows[2]

[1]Howard Hughes Medical Institute/Janelia Research Campus, Ashburn, United States; [2]Department of Zoology, University of Cambridge, Cambridge, United Kingdom

**Abstract** Jumping in planthopper and froghopper insects is propelled by a catapult-like mechanism requiring mechanical storage of energy and its quick release to accelerate the hind legs rapidly. To understand the functional biomechanics involved in these challenging movements, the internal skeleton, tendons and muscles involved were reconstructed in 3-D from confocal scans in unprecedented detail. Energy to power jumping was generated by slow contractions of hind leg depressor muscles and then stored by bending specialised elements of the thoracic skeleton that are composites of the rubbery protein resilin sandwiched between layers of harder cuticle with air-filled tunnels reducing mass. The images showed that the lever arm of the power-producing muscle changed in magnitude during jumping, but at all joint angles would cause depression, suggesting a mechanism by which the stored energy is released. This methodological approach illuminates how miniaturized components interact and function in complex and rapid movements of small animals.

\*For correspondence:
siwanowiczi@janelia.hhmi.org

**Competing interests:** The authors declare that no competing interests exist.

## Introduction

Jumping is one of the fastest and most powerful movements generated by animals. Consequently it places great demands on the nervous system, muscles and the skeleton to ensure that it can be performed successfully and repeatedly. These movements are frequently used to escape from predators so that the cost of failure is high. In insects different jumping mechanisms have evolved independently several times, but all that use the legs for propulsion fall into two broad types. First, contractions of the muscles act directly on long levers. Examples of insects which use this mechanism are bush crickets, the propulsive hind legs of which can be up to 2.6 times the body length (*Burrows and Morris, 2003*). The length of the levers enables them to generate the necessary leverage, but take 20–30 ms to be moved to full extension and accelerate the insect to take-off.

In the second jumping mechanism, the propulsive legs are short relative the body and deliver their power in a few milliseconds by using a catapult-like mechanism (*Bennet-Clark, 1975a1975*; *Bennet-Clark and Lucey, 1967*). The champion jumping insects, as judged by their take-off velocities, all use a catapult mechanism and belong to two groups of the true bugs (Hemiptera), the closely related planthoppers (Fulgoroidea) and froghoppers or spittle bugs (Cercopoidea). The refinements of the body for this mechanism are thus likely to be developed the most in these insects. The adults of both may live in similar habitats feeding on the sap of plants which they suck with piercing mouthparts. They jump prodigiously propelled by their hind legs. Planthoppers nymphs live freely alongside the adults and are also very able jumpers. By contrast, the nymphs of froghoppers do not jump and instead live a rather restricted life either underground, or on plant leaves,

surrounding themselves with a protective froth which they make by blowing air into their urine. The adults generate take-off velocities ranging from 4 to 5.8 m s$^{-1}$ in their best jumps (*Burrows, 2006a*, *2009*, *2014*). They achieve these remarkable velocities by accelerating their bodies in about 1 ms and in so doing experience forces that can reach 700 g; most humans pass out with forces 100 times less. The power needed to generate such rapid and powerful movements can be 1000 times greater than the contractile limits of normal muscle (*Askew and Marsh, 2002*; *Ellington, 1985*; *Josephson, 1993*; *Weis-Fogh and Alexander, 1977*). A complex motor pattern is fashioned by the nervous system and delivered to the muscles so that they build up the required force slowly and without moving the two propulsive hind legs (*Burrows, 2007*; *Burrows and Bräunig, 2010*). The energy of these slow contractions is stored in mechanical distortions of specialised cuticular structures of the thoracic skeleton associated only with the hind legs (*Burrows et al., 2008*). The energy is then released suddenly, by mechanisms that are not yet fully understood, to generate rapid propulsive movements of the hind legs. These energy storage devices must be able to deliver the necessary power repeatedly and reliably without fracturing. The insects must also ensure that the starting point for the movements is always the same so that they can perform jumps with predictable outcomes (*Brackenbury, 1996*). This requires that the stores must be both strong and elastic, properties that come from their constituent composite materials, hard cuticle and cuticle containing resilin (*Burrows et al., 2008*).

The elastic protein resilin was initially discovered in the tendons of flight muscles that must reliably generate many repetitive cycles of movement during the lifetime of an insect (*Weis-Fogh, 1960*), but has since been found in many different places in the cuticle of arthropods. In particular, it is associated with energy storage devices in a range of insects from fleas (*Bennet-Clark and Lucey, 1967*), froghoppers (*Burrows et al., 2008*) and planthoppers (*Burrows, 2010*). The latter two examples have some of the largest volumes of resilin relative to body size in any insect.

The great speed and power of the jumping movements also requires close interactions between the neurons, muscles and the skeleton. This is particularly important in synchronising the movements of the two propulsive legs to within 30 μs of each other in planthoppers (*Burrows, 2009*). Without this synchrony, the body will spin with little accompanying forward movement, as much of the mechanical energy acts to rotate the body. Such levels of synchronisation are beyond the limits of neural control but are accomplished by mechanical means such as the interactions between functional gear wheels on each propulsive hind leg of nymphal planthoppers (*Burrows and Sutton, 2013*). These enmeshed gears ensure that when one hind leg moves, the other also moves at the same time.

To understand how these fast jumping movements are generated, we have reconstructed from confocal scans the 3-D structure of three elements of the jumping mechanism to address three questions. First, what is the structure of the thorax that enables the energy generated by the muscle contractions to be stored and power a jump? Second, what is the distribution of resilin in these structures? Third, how do the muscles and their mechanical advantages change during jumping movements? We then analysed different species of planthoppers and froghoppers to seek common features or differences in these mechanisms. This approach has given insights into the functional design of the energy stores, how energy is stored and then released, and how the rapid propulsive movements of the hind legs are generated.

## Results

### General thoracic structure for jumping in planthoppers

The adults and nymphs (*Figure 1A*) of planthoppers are all adept at jumping, propelled by the synchronous, rapid depression movements of the two hind legs. The front and middle legs set the attitude of the body but apparently contribute little to propulsion. The metathorax of both adults (*Figure 1B,C*) and nymphs (*Figure 1D,E*) is reinforced internally by large, paired pleural arches (apodemes) that are substantial and complex structures in comparison to those in the other thoracic segments. From their ventral articulations with the two hind coxae, these arches run anteriorly and dorsally to articulate with the hinges of the hind wings. In adults (*Figure 1B,C*) each enlarged pleural arch forms a tight linkage with an anterior and dorsal extension of a hind coxa. By contrast, a pleural

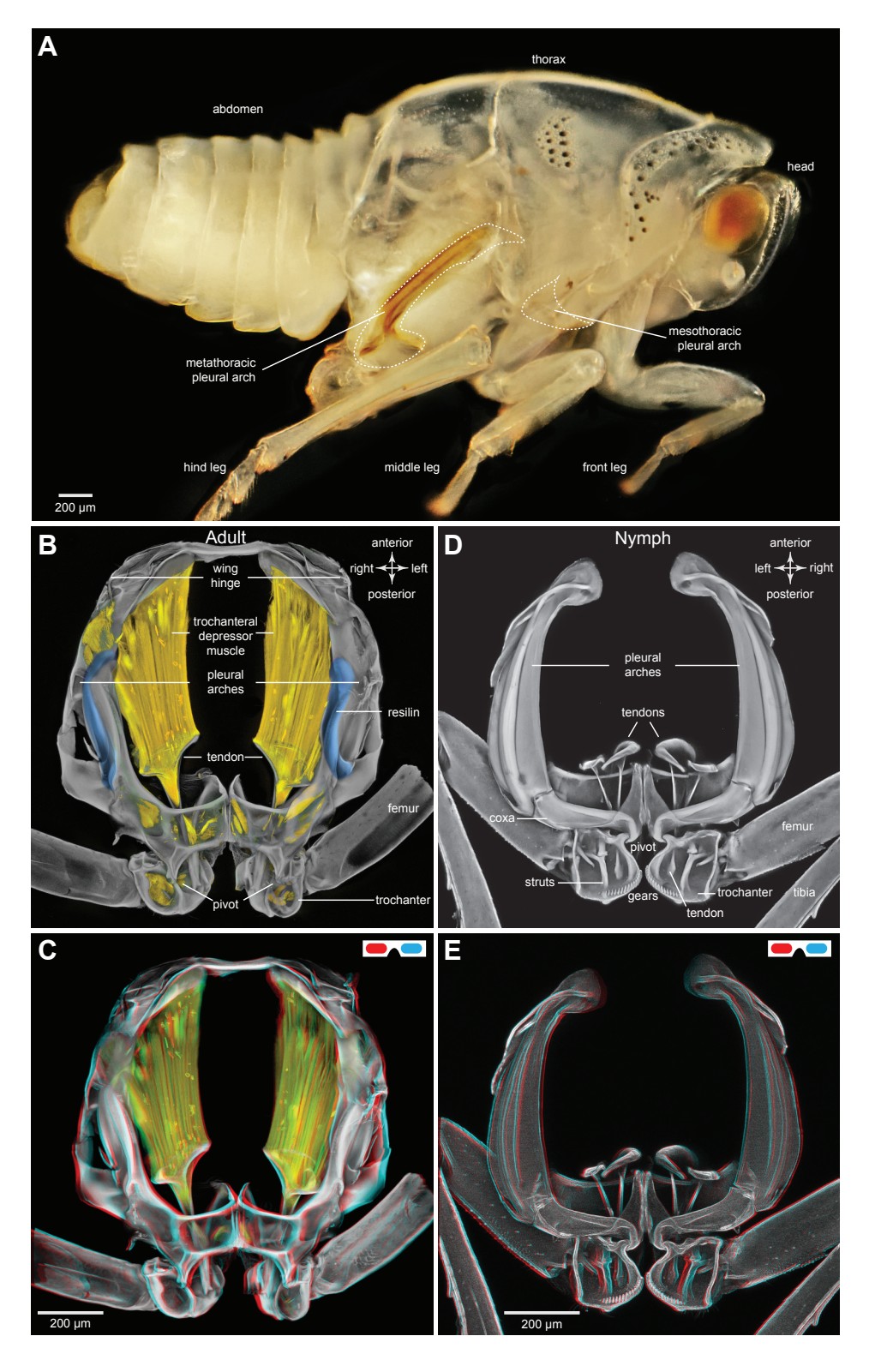

**Figure 1.** Structure of the skeletal elements in the metathorax of planthoppers associated with jumping. (**A**) Side view of an intact nymph of *Metcalfa pruinosa*. The right pleural arch of the metathoracic segment is visible through the external cuticle. (**B**) Two dimensional ventral view of the thorax of an adult Delphacid planthopper, *Stenocranus acutus*. The large trochanteral depressor muscle (M133b/c) and parts of other muscles are shown in yellow and resilin in the pleural arches is shown in blue. All other soft tissue was removed. (**C**) Three dimensional rendering (indicated by the symbol of

*Figure 1 continued on next page*

*Figure 1 continued*

coloured glasses) of the same ventral view of the adult. Muscle M133b/c is again shown in yellow. The scale bar in (C) also refers to (B). (D) Two dimensional dorsal view of the thorax of a nymph of *S. acutus*. E. Three dimensional ventral view of the nymph. The scale bar in (E) also refers to (D). In (D,E) all soft tissue in the thorax was digested away to leave only the skeletal structures. Some cuticle on each coxa was also removed to reveal the articulation of a coxa with a pleural arch on each side.

arch in a nymph has a more blade-like structure and its articulation with a coxa is clearly delineated by a flange that allows some movement between the two elements (*Figure 1D,E*). In the images of the nymphs, the lateral and dorsal outer layers of thoracic cuticle and the dorsal walls of the coxae have been removed to show these features more clearly.

Part of the ventral surface of the metathorax in both adults and nymphs is made of flexible, transparent membrane through which the large, paired trochanteral depressor muscles are visible (*Figure 1B,C*). These muscles fill most of the internal thoracic space, and power the rapid movements of the hind legs in jumping. In nymphs the anterior origins of these paired muscles are continuous with and a part of metathoracic pleural arches. The mesothoracic pleural arches form a support for the middle legs and do not contribute to the structures of the jumping apparatus of the hind legs (*Figure 2A* inset). In adults, however, the anterior ends of the metathoracic pleural arches articulate laterally with flattened expansions of the mesothoracic arches to form the origins of the trochanteral depressor muscles (*Figure 2B–D*. For each Figure shown in 3D, we provide a 2D version). Each trochanteral depressor muscle then inserts onto a prominent funnel-shaped tendon in the thorax that tapers to a stout rod as it passes through a coxa to insert on the medial wall of a trochanter (*Figure 1B–D*).

The paired trochanteral depressor muscles (M133b/c shown in yellow in *Figure 1B,C*) contract before a jump but the hind legs remain stationary in their fully cocked (levated) positions (*Burrows, 2007*; *Burrows and Bräunig, 2010*). This action pulls on the trochantera with the result that the pleural arches are bent and their curvature increased (*Video 1*, Supplementary material). The positions of the most anterior articulations of the pleural arches with the wing hinges stay constant but the posterior articulations with the coxae, and the coxae themselves, move forwards as a unit. Then, as the pleural arches unfurl, the coxae move rapidly backwards and the hind legs straighten by depression at the coxo-trochanteral joints and extension at the femoro-tibial joints to propel the jump. The coxae thus form a stable base which allows cocking (levation) and propulsive (depression) movements of the trochantera about their dorsal and ventral articulations with the coxae. This stability enables the femora and tibiae to straighten and transmit the propulsive forces for a jump to the ground through the tarsi.

To withstand these forces, a hind trochanter is reinforced by internal struts (*Figure 1D*) and its medial and lateral walls are thickened. In nymphs, but not adults, the external medial surface of a trochanter has a gear wheel (*Figure 1D,E*). The gears on each trochanter intermesh during cocking movements in preparation for a jump (levation) and during the propulsive jumping movements themselves (depression) of the hind legs (*Burrows and Sutton, 2013*). These features of the pleural arches, thoracic structures and proximal leg joints described above are also common to the five other families of planthoppers analysed with some differences in relative sizes (*Figure 2Ai–v*).

## Distribution of resilin in the energy stores of adult planthoppers

An integral material element of the metathoracic energy stores (pleural arches) is the rubber-like protein resilin as revealed by staining with acridine orange, but shown in blue (*Figures 1B* and *3*, *Table 1*). In a ventral view of the right pleural arch of an Issid planthopper, the resilin is seen as an elongated region 500 µm wide, 200 µm thick and 1250 µm long which curves laterally at its posterior end near its articulation with the coxa (*Figure 3Ai,ii*; *Table 2*). In sections the resilin is surrounded on three sides by cuticle of a pleural arch which contains little or no resilin. On the lateral surface the cuticle is permeated by two antero-posterior longitudinal holes (*Figure 3Aii*, white arrowheads in sections 2,3). The almost complete encirclement by cuticle means that in a dorsal view of a pleural arch, most of this resilin is hidden with only a small part protruding anteriorly close to the articulation with the wing hinge and the site of origin of the trochanteral depressor muscle (*Figure 3Aiii*). From these images, the volume of resilin in a pleural arch was calculated to be 0.138 mm$^3$. The ratio of the

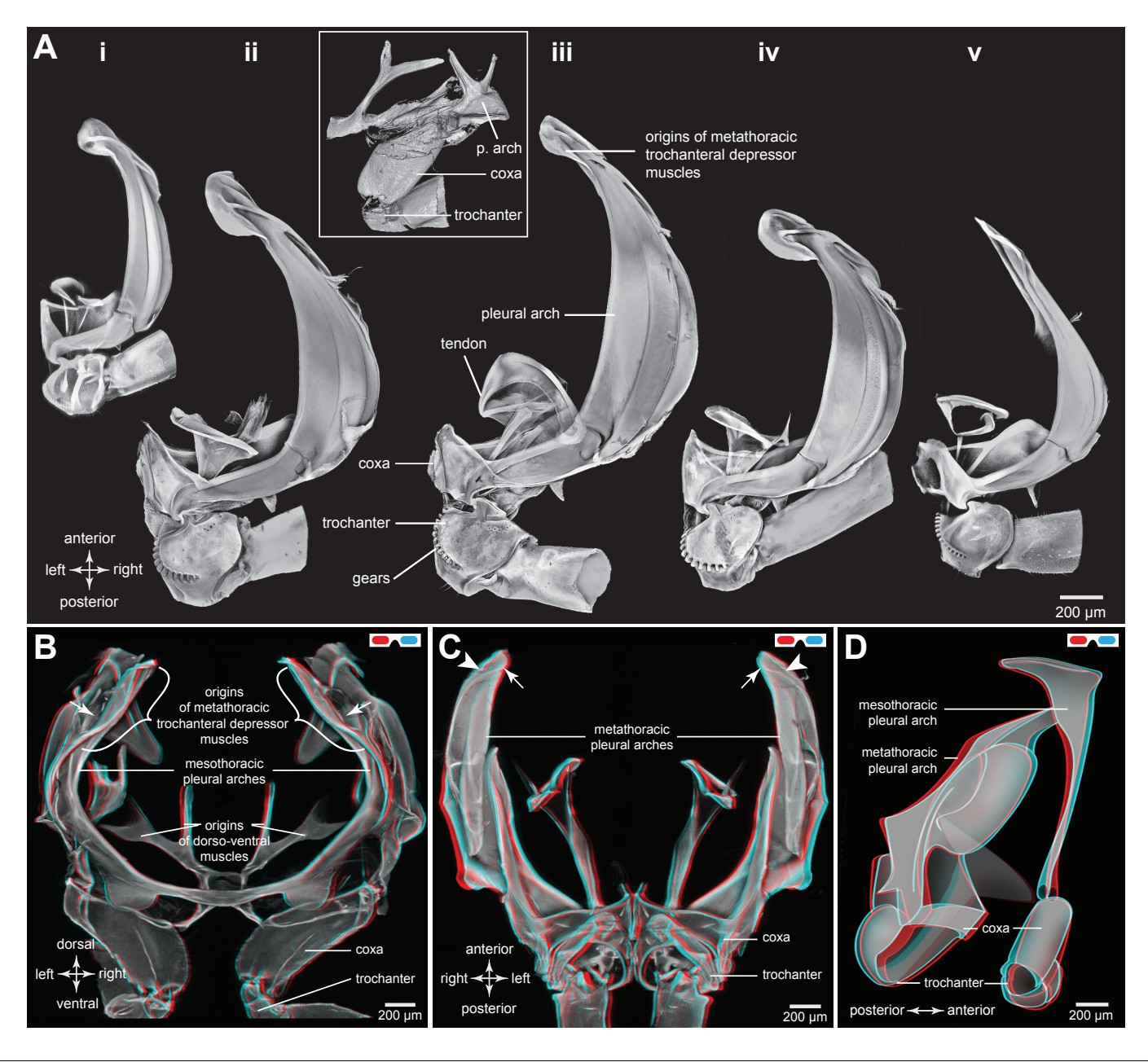

**Figure 2.** Comparison of the structure of the pleural arches in nymphs of five families of planthoppers. (**Ai**) Delphacidae, *Stenocranus acutus*. (**Aii**) Issidae, *Thionia bullata*. (**Aiii**) Acanaloniidae, *Acanalonia conica*. (**Aiv**) Flatidae, *Metcalfa pruinosa*. (**Av**) Caliscelidae, *Bruchomorpha oculata*. In each of these nymphs the right pleural arch is viewed dorsally to show its articulation with the coxa, the tendon of the trochanteral muscle, and the gears on the trochanter. Inset: mesothoracic struts of *A. conica* that give rise to the mesothoracic pleural arch in the adult. (**B**) Mesothoracic arches of an adult *Flatormenis proxima* viewed dorsally, imaged using red autofluorescence (**C**). Metathoracic arches of *F. proxima* viewed ventrally. Arrows denote articulation sites of the arches; arrowheads, articulations with a wing hinge. (**D**) Schematic representation based on confocal data showing relative the positions of the meso- and metathoracic pleural arches viewed medially.

The following figure supplement is available for figure 2:

**Figure supplement 1.** Comparison of the structure of the meso- and metathoracic pleural arches in an adult planthopper.

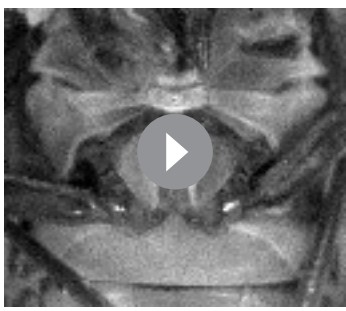

**Video 1.** Bending of the pleural arches of a planthopper during natural jumping.     High speed movie taken at 30 000 frames per second and with an exposure time of 0.03 ms of the jumping movements of an adult planthopper *Issus coleoptratus*. Before a jump, the hind legs are moved into their fully levated position so that the coxae are located anteriorly within the metathorax. The large trochanteral depressor muscles then contract without moving the hind legs, resulting in the pleural arches being bent. As the jump is released, the pleural arches unfurl and both coxae move backwards rapidly as a unit. The more distal parts of the two hind legs also move rapidly in unison.

volume of resilin in both pleural arches relative to body mass was 12.82 mm$^3$ g$^{-1}$ (*Table 2*). In adult planthoppers, resilin was also found in a medial protrusion of a hind trochanter which pressed against a similar protrusion from the other trochanter when the hind legs were cocked (levated) in readiness for jumping (*Burrows, 2010*) (*Figure 3Aiii*).

A similar pattern to the distribution of resilin was also seen in other families of planthoppers. For example, in Derbid planthoppers the region of resilin was somewhat shorter but broader and the lateral cuticle of the pleural arch was not hollowed, but all other features were similar (*Figure 3Bi–iii*). In three dimensional views of a pleural arch of both Acanaloniid planthoppers and Derbid planthoppers, the region containing resilin was encased by medial and lateral plates of cuticle (*Figure 4A,B*). This surrounding cuticle was reinforced laterally with antero-posterior folds while its medial surface was thinner and flatter. An apparent oval hole that was normally covered by a thin cuticular membrane was revealed in both families toward the lateral edge of the arch at its articulation with the coxa (*Figures 3A, B* and *4A,B*).

## Bending of the energy stores of adult planthoppers

To simulate the bending movements of the pleural arches that have been observed during the preparation for a jump, isolated thoraces were gently squeezed between two coverslips by applying increasing forces while scans were made of any resulting movements of an arch (*Figure 5A*). The forces acted in the same direction as would be experienced during a natural jump. Successive images were then linked together into a movie (*Video 2*, Supplementary material) that allowed the displacements of an arch to be determined. From such movies the main bending region was identified as in the middle of the arch (*Figure 5A–C*). Here, and close to the hole, a small tendon inserts on the inner dorsal part of the pleural arch and runs anteriorly (*Figure 5B,C*). There are also two curved cuticular parts of the pleural arch, one lateral and one more medial, that will be subject to the maximal bending forces, suggesting that they might act as springs in additional to those in the pleural arches (*Figure 5B,C*).

## Muscle tendons in planthoppers

The tendons of the trochanteral depressor muscles that generate the power for jumping have a complex funnel shape, made of cuticle in both adults and nymphs (*Figure 6A–D*). Through the transparent ventral cuticle of the thorax, the tendons of these bilateral muscles could be seen to move anteriorly in unison within the metathorax as the hind legs were depressed to propel a jump, and then posteriorly as the legs were levated in preparation for the next anticipated jump. The rim of cuticle that forms the mouth of the funnel is not a complete circle and this shape suggests that when the muscle contracts it may be distorted and thus store additional energy. More posteriorly, the tendon tapers markedly so that as it passes through the coxa it is a narrower and rigid, rod-like structure (*Figure 6B,C*). In specific places the wall of this narrow part of the tendon is indented at the insertion point of a group of fibres that constitute a small part (M133a) of the trochanteral depressor muscle (*Figure 6D*). The tendon then flares into broader but flexible membrane at its insertion on the medial ventral wall of the trochanter (*Figure 6D*).

To store energy in advance of a jump a trochanteral depressor muscle (*Figure 7A,B*) must contract without depressing the trochanter. A possible mechanism by which this might be achieved is

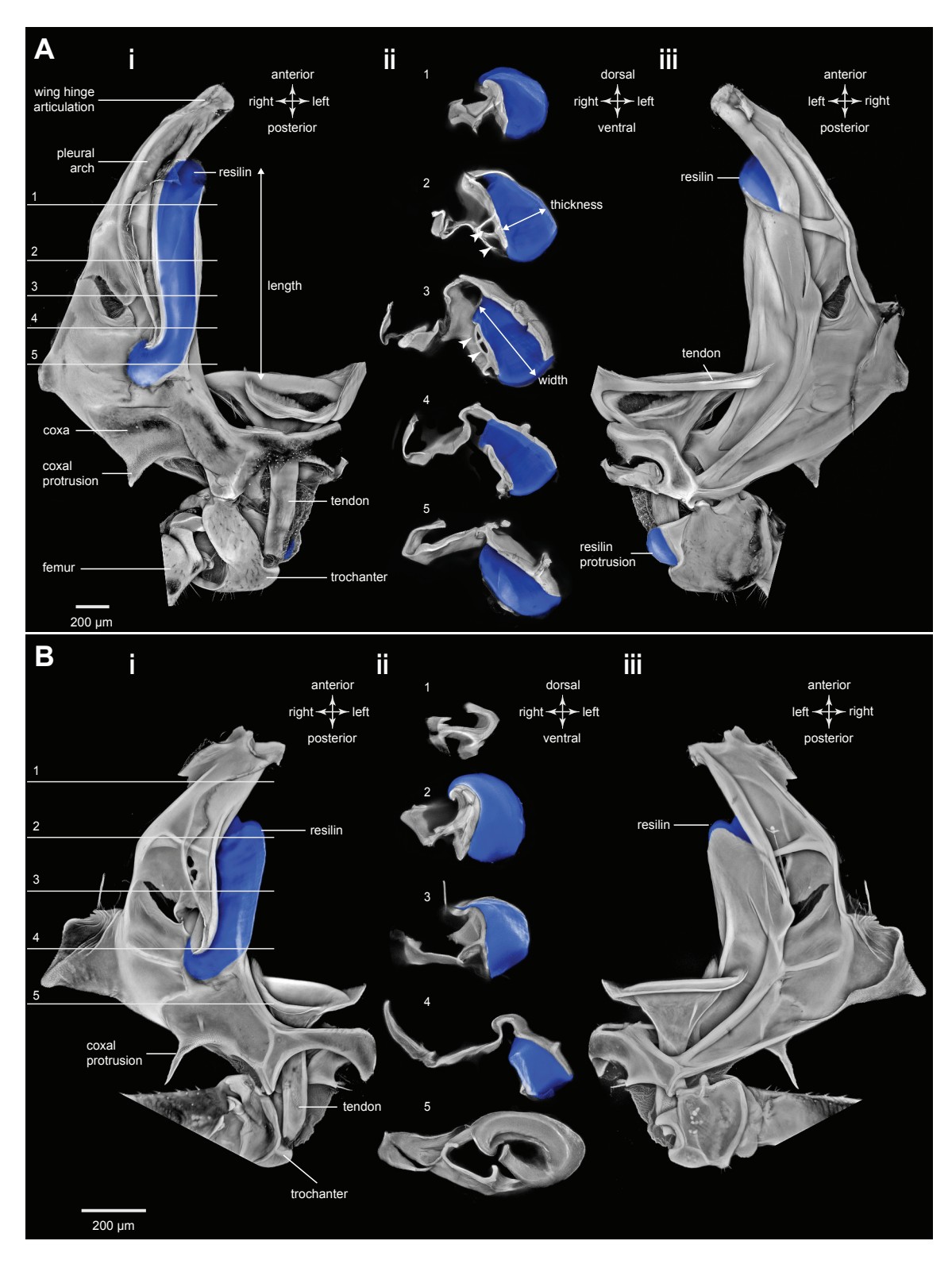

**Figure 3.** Resilin in the thorax of adult planthoppers. (**A**) The skeletal elements of the right side of the metathorax of the Issid planthopper *Thionia bullata* after all soft tissue and soft cuticle was removed. The resilin, visualized using Acridine Orange staining, is shown in blue. (**Ai**) Ventral view. (**Aii**) Sections cut at the five levels indicated in (Ai). The white arrowheads in sections 2 and 3 indicate the holes. (**Aiii**) Dorsal view. (**B**) The same skeletal

*Figure 3 continued on next page*

*Figure 3 continued*
elements of the right thorax of the Derbid planthopper *Apache degeeri*. (**Bi**) Ventral view. (**Bii**) Sections cut at the five levels indicated in (**Bi**). (**Biii**) Dorsal view.
The following figure supplement is available for figure 3:

**Figure supplement 1.** Comparison of methods for resilin visualization in planthopper and froghopper energy stores.

for the lever arm of the tendon to change in sign relative to the pivot of the coxo-trochanteral joint (*Gorb, 2004a2004*). In this way, when the hind leg is fully cocked in readiness for a jump, the tendon would be to one side of the pivot axis so that the action of the muscle would be to levate rather than depress the leg. If another muscle were to move the loaded tendon to the other side of the pivot axis, the action of the depressor muscle would then be to depress the trochanter and propel a jump. To determine whether the lever arm of the depressor muscle does reverse in this way relative to the pivot axis, the images making up the three dimensional stacks were analysed individually and the positions of the tendon and the ventral and dorsal pivots of the joint were marked. This enabled the lever arm of the depressor tendon to be determined relative to the axis of rotation of the trochanter about the coxa (lines drawn in *Figure 7B,C*). The whole process was then repeated with the trochanter held at different angles relative to the coxa throughout its whole range of levation and depression movements. Two examples are shown; first with the trochanter almost fully depressed about the coxa (*Figure 7B*); second with the trochanter fully levated (cocked) (*Figure 7C*). When the joint was fully depressed the lever arm of the tendon was at it furthest (medial) position relative to the axis of rotation of the joint and thus had its greatest mechanical advantage (*Figure 7B*). At its most levated position, the lever arm was close to the axis of rotation of the joint but was still medial to it (*Figure 7C*). The lever arm thus became smaller as the joint was levated but its sign always remained the same. Thus throughout the entire range of trochanteral movements, the depressor muscle always acted to depress the trochanter.

## Muscles in planthoppers

The main body of the trochanteral depressor muscle (M133b/c) in the metathorax is very much larger than the equivalent muscles in the other two thoracic segments (*Figure 7A*). The fibres of the main body insert on the rim of the tendon, with some central fibres extending further to insert on the interior of the tapering region of the tendon (*Figures 6A* and *7B,C*; *Video 3*, Supplementary material). A small part of the depressor muscle (M133a) originates on the medial wall of the coxa and inserts on the straight part of the tendon within the coxa itself (*Figure 7B,C*). When viewed at higher magnification the insertions on the tendon are made by an array of fibrils similar in number to the constituent muscle fibres (*Figure 7D,E*). As the trochanter rotates from its fully levated position to a fully depressed one, the orientation of the fibres of this small muscle changes markedly in accord with the movements of its insertion site on the main tendon and the fixed origin of the fibres on the coxal wall (*Figure 7D,E*).

**Table 1.** Summary of the stains and the excitation and emission wavelengths used when imaging specimens.

| Tissue | Staining | Excitation wavelengths, nm | Emission wavelengths, nm |
|---|---|---|---|
| Chitin | Autofluorescence | 561 | 568–683 |
| Chitin and Muscles (nymphs) | Calcofluor White | 405 | 410–570 |
| | Texas Red Phalloidin | 594 | 599–660 |
| Chitin and Muscles (adults) | Autofluorescence | 405 | 410–465 |
| | Texas Red Phalloidin | 594 | 599–660 |
| Chitin and Resilin | Autofluorescence | 405 | 410–465 |
| | Acridine Orange | 488 | 520–580 |

**Table 2.** Volumes of resilin in a pleural arch, and ratios of both arches relative to body mass in the planthopper *Thionia bulllata* and the froghopper *Lepyronia quadrangularis*. See **Figure 3Ai**, ii for definitions of width, thickness and length and Figure 9A–C for definitions of the two resilin regions in froghoppers.

| | Width, μm | Thickness, μm | Length, μm | Volume one arch, mm$^3$ | Ratio, both arches: body mass, mm$^3$ g$^{-1}$ |
|---|---|---|---|---|---|
| Planthopper | 500 | 200 | 1250 | 0.138 | 12.8 |
| Froghopper | | | | | |
| Large region | 710 | 290 | 1450 | 0.121 | |
| Ovoid region | 440 | 150 | 710 | 0.044 | |
| Total | | | | 0.165 | 18.7 |

The two levator muscles are much smaller than the whole depressor muscle; the fibres of the larger levator muscle (M132) originate on a lateral, anterior and dorsal coxal process fused with the pleural arch, while the smaller levator muscle (M131) originates on the ventral coxal wall (**Figure 7B, C**). Their tendons insert on the anterior wall of the trochanter.

## General thoracic structure of froghoppers

Nymphal froghoppers do not jump and their metathorax lacks the enlarged muscles that power the movements of the hind legs in jumping (**Figure 8A**). The nymphs also lack functional wings and do not fly. They also lack the large thoracic muscles that would power flying. By contrast, the metathorax of adult froghoppers is dominated by the pair of large trochanteral depressor muscles that propel jumping and the mesothorax by large muscles that power flying (**Figure 8B**). The origins of the froghopper trochanteral depressor muscles are larger and more ventral than in planthoppers. Enlarged pleural arches extend from the coxae to the articulation of the hind wings (**Figure 8C**), but the articulation of each coxa with a pleural arch allows rotation through a greater range of angles. As in adult planthoppers, the origins of the trochanteral depressor muscles include part of the mesothoracic pleural arches. These origins run dorso-ventrally as a thin curved sheet of cuticle and serve the dual function of supporting the metathoracic trochanteral muscles posteriorly and the mesothoracic flight muscles anteriorly (**Figure 8B,C**). Each coxa has a prominent external protrusion that projects laterally and posteriorly and which is covered in numerous much smaller protrusions (microtrichia) (**Figure 9A,D**). The ventral wall of the metathorax is made of opaque cuticle so that the underlying jumping muscles are not visible from the outside.

## Distribution of resilin in the energy stores of adult froghoppers

Froghoppers have two separate regions of resilin associated with each metathoracic pleural arch (**Figure 9**). The large region, which is comparable in its location to the single region in planthoppers, measures 710 μm wide x 290 μm thick x 1450 μm long at its maximum dimensions in *Lepyronia quadrangularis* (**Table 2**), and is sandwiched between lateral and medial plates of cuticle of a pleural arch (**Figures 9A,B** and **10A**). At the anterior end the cuticular plates are thrown into a series of folds, one long and one shorter, which would be expected to stiffen the overall structure (**Figure 9B, C**). The second smaller and ovoid region of resilin overlaps with the larger region but extends more posteriorly and toward the articulation with a coxa. This region, which is not present in planthoppers, measures 440 μm wide x 150 μm thick x 710 μm long at its maximum dimensions (**Figures 9B,C** and **10B**). Posteriorly, where these two regions of resilin overlap, they are separated from each other by a thin layer of cuticle and are bounded dorsally, ventrally and medially by plates of cuticle. From these images the volume of resilin in a froghopper pleural arch was calculated to be 0.165 mm$^3$ made up from a volume of 0.121 mm$^3$ in the larger region and 0.044 mm$^3$ in the smaller ovoid region (**Table 2**). The ratio of the volume of resilin in both pleural arches relative to body mass was 18.74 mm$^3$ g$^{-1}$.

Anterior to a coxal protrusion on either side of the thorax, the external cuticle is invaginated to form an entrance (curved white arrows in **Figure 9A,C,D**) to a tunnel within the pleural arch. This extends as far anteriorly as the start of the groove and the anterior part of the ovoid region of resilin.

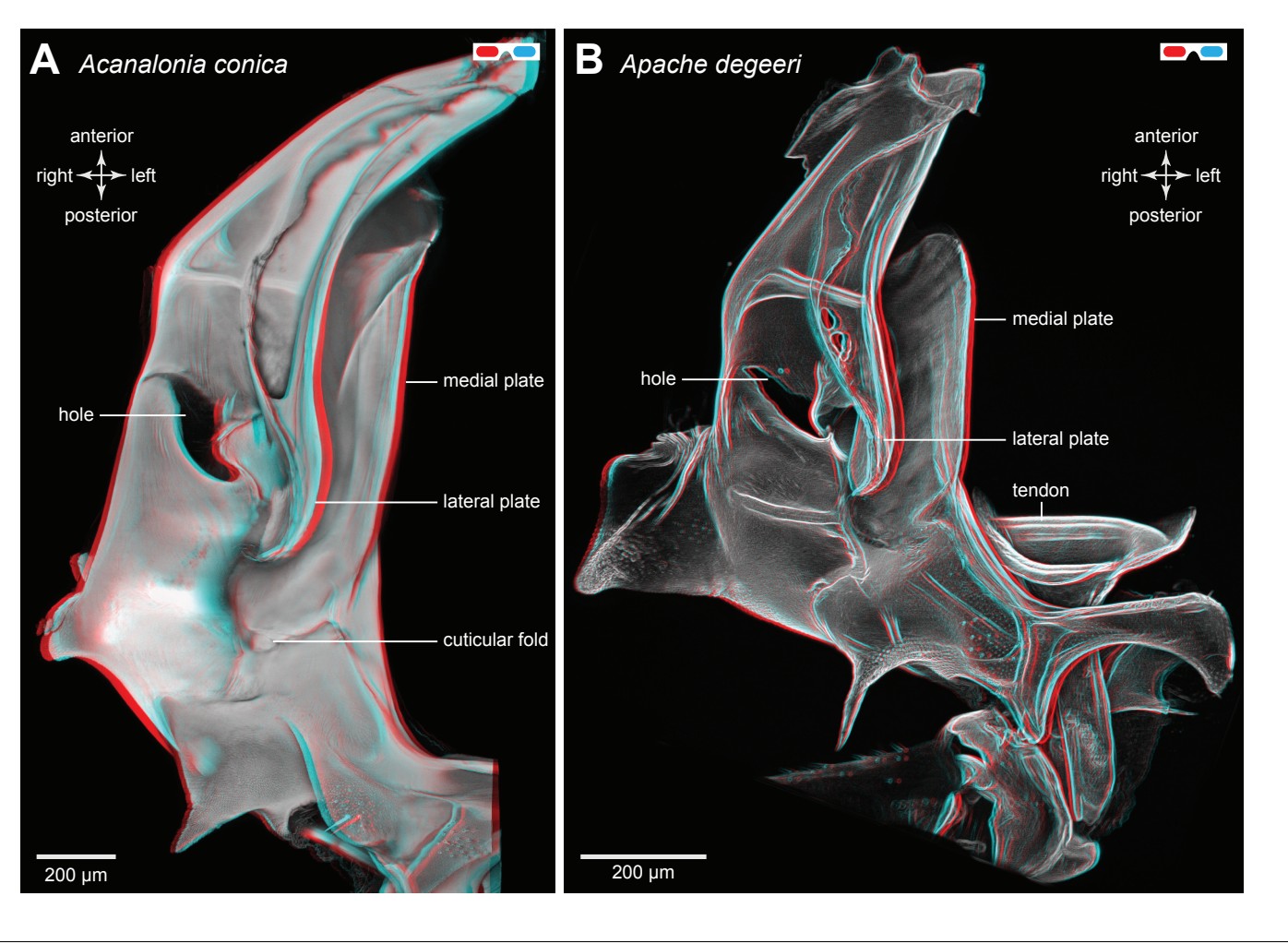

**Figure 4.** Three dimensional structure of a pleural arch. (A) The right pleural arch of the Acanaloniid planthopper *Acanalonia conica* viewed ventrally. The lateral plate of the arch is reinforced by antero-posterior infoldings, and the medial plate is thin but curved. The region between these two plates is normally filled with resilin. (B) The right pleural arch of the Derbid *Apache degeeri* viewed ventrally and in which edges have been emphasised (using 'find edges' filter in ImageJ). Again the region delineated by the two plates is normally filled with resilin. The arches were imaged using red autofluorescence.

The following figure supplement is available for figure 4:

**Figure supplement 1.** Structure of a pleural arch.

This tunnel is thus open to the outside and provides an air-filled space (asterisks in *Figure 9B*) alongside the resilin and cuticle for about 40% of the length of a pleural arch (*Figure 10A,B*).

## Tendons and muscles in adult froghoppers

The tendon of the trochanteral depressor muscle (M133), which also powers the jump in froghoppers, has a different shape to that in planthoppers (*Figure 11A,B*). The muscle itself consists of four parts, the two larger parts (M133b/c) of which fill most of the space in the metathorax and two smaller parts (M133a, M133d) which attach to the tendon in the coxa (*Figure 12A,B*; *Video 4*, Supplementary material).

The insertion site for the main medial bundle of fibres that constitute M133c is a surface with edges that curve anteriorly. Close to these insertions and still within the thoracic cavity, the tendon branches into a thinner lateral and anteriorly directed part to which the pinnately arranged fibres of

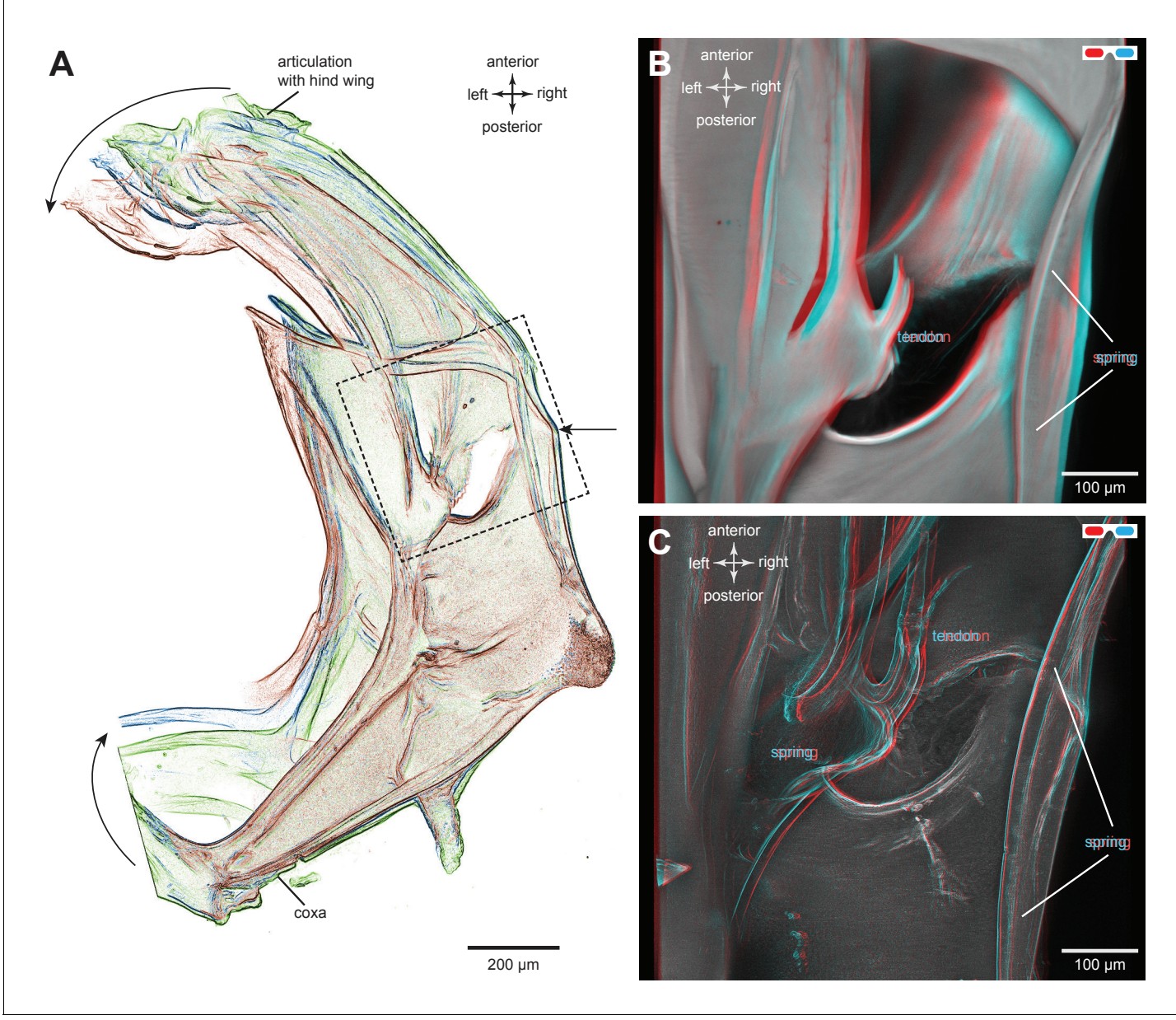

**Figure 5.** Structures associated with the bending of the pleural arches in planthoppers. (A) Outlines of the right pleural arch viewed dorsally of the Flatid planthopper *Flatormenis proxima* taken from three individual frames of a movie (Supplementary material, *Video 2*) whilst a bending force was applied. The straight arrow points to the position of the region about which bending occurred and the curved arrows indicate the direction of bend at the anterior and posterior ends. (B) An expanded dorsal view of the area in (A) marked by the dashed box, showing the insertion of a tendon from a small muscle and the hinge region of the pleural arch. (C) The same image in which edges are emphasised to reveal a potential spring underneath the tendon insertion and a second potential curved spring on the right edge. These images can be seen in 3-D rotational views in Supplementary Material, *Video 3*.

The following figure supplement is available for figure 5:

**Figure supplement 1.** Structures associated with the bending of the pleural arches in planthoppers.

M133b insert along its length. As the combined tendon enters the coxa it tapers gradually to assume a dorso-ventrally flattened shape. Along its medial side is the insertion site of a small part of the muscle (M133a) that originates on the ventral medial wall of the coxa (*Figure 12A*). As in planthoppers, the tendon flares and becomes more flexible close to its insertion on the medial wall of the

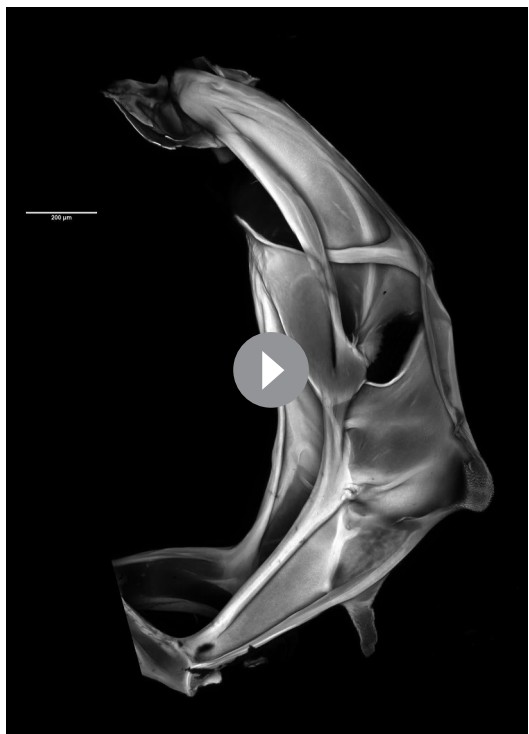

**Video 2.** Imposed bending movements of a pleural arch of a planthopper.      The isolated pleural arch of the planthopper *Flatormenis proxima* was restrained between coverslips and a sustained force was applied to the top coverslip while a scan was made. The movie consists of three frames showing the pleural arch in a relaxed state and two compressed conformations.

trochanter. M133d is an even smaller bundle of longer fibres that originates in the ventral thorax just lateral and posterior to the metathoracic ganglion and inserts on the tendon within the coxa (*Figure 12B*).

## Discussion

Three dimensional reconstruction of images of the skeletal structures, the muscles and their tendons that generate the rapid propulsive movements of the hind legs, have revealed three features that meet the extraordinary mechanical demands of jumping. **First**, the substantially enlarged pleural arches act as energy stores for jumping and provide the necessary rigidity to withstand the forces that are generated. **Second**, large volumes of resilin are intimately bonded in complex ways with regions of the pleural arches that apparently contain no resilin. These composite structures store and release energy to power repeated jumping. **Third**, large, paired trochanteral depressor muscles occupy most of the internal volume of the metathorax. The changing lever arms of these muscles suggest a possible mechanism for the release of the stored energy to power a jump. By contrast, the other two pairs of legs which do not contribute propulsion for jumping have smaller pleural arches without resilin and smaller trochanteral depressor muscles.

### Structure of energy stores

In nymphal planthoppers and adult planthoppers and froghoppers, the metathoracic pleural arches are curved bow-like structures with folds and grooves that apparently give increased strength. When the large trochanteral depressor muscles contract slowly before a jump, the pleural arches are bent like archery bows and store elastic energy, but the legs do not move. Sudden release of this stored energy powers the rapid movements of the hind legs in a catapult-like action and drives the insect to a high take-off velocity.

The metathoracic pleural arches of all six families of planthoppers analysed had similar common features. A region was identified in the middle of a pleural arch close to where a small laterally placed muscle tendon attaches and two small curved plates also occur. At this point these plates are likely to be subject to the greatest bending forces and might therefore act as additional springs. In planthoppers longitudinal holes are present in the pleural arches and in froghoppers an invagination of the external cuticle forms an air-filled tunnel that extends for some 40% of their length of each arch alongside the cuticular wrapped regions of resilin. These spaces may be a consequence of developmental processes, but functionally they should reduce the mass of the arches and their juxtaposition to the resilin suggests that they may allow easier bending and room for local bulging during energy storage.

### Resilin and energy stores

The protein resilin has previously been detected by its characteristic blue fluorescence under specific wavelengths of UV light (*Andersen, 1963*; *Andersen and Weis-Fogh, 1964*; *Malencik et al., 1996*), the dependence of the fluorescence on pH (*Neff et al., 2000*), its staining with very dilute solutions of Methylene Blue or Toluidine Blue (*Weis-Fogh, 1960*; *Young and Bennet-Clark, 1995*) and immunostaining with an anti-Rec1 resilin polyclonal antibody (*Elvin et al., 2005*; *Burrows et al., 2011*;

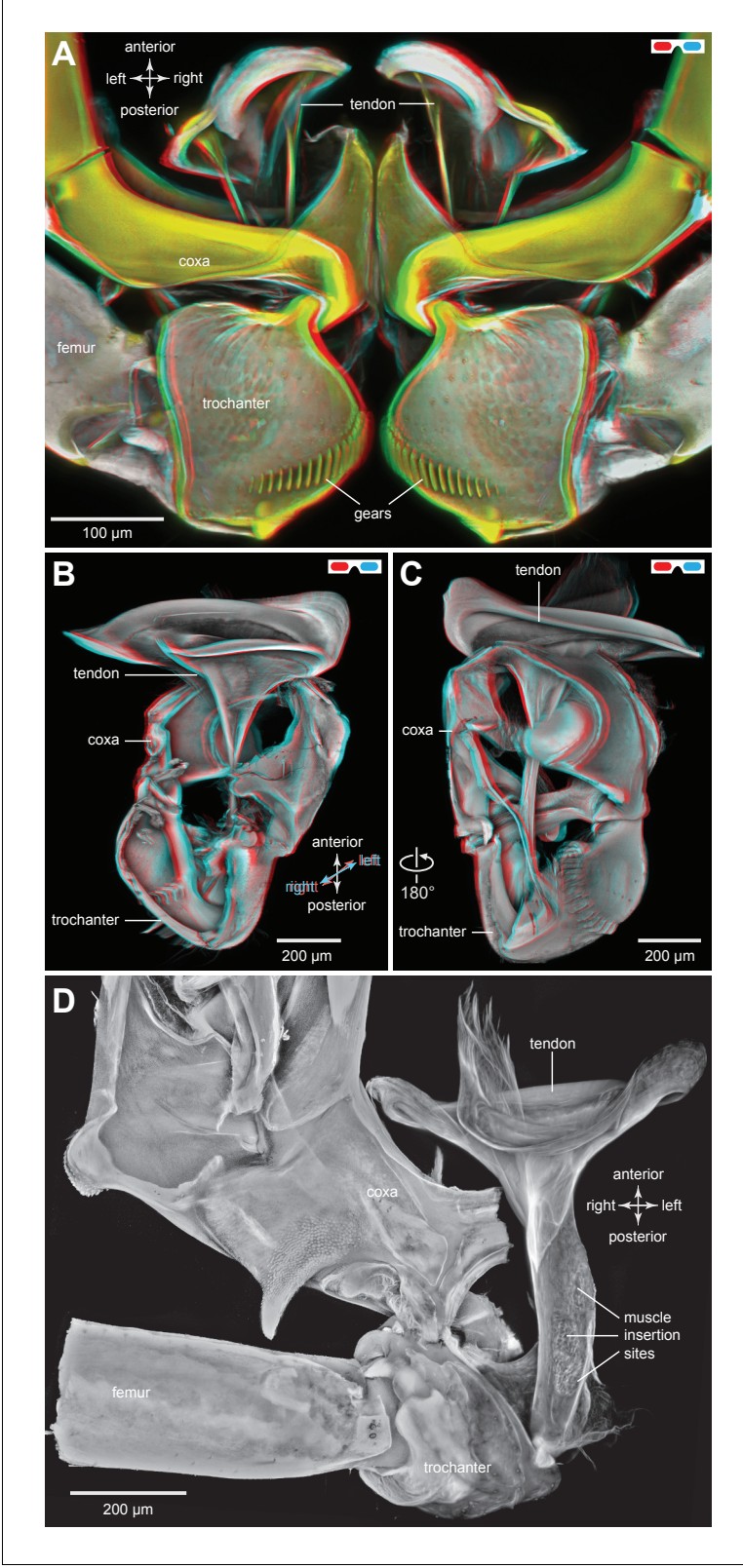

**Figure 6.** The tendon of the trochanteral depressor muscle in planthoppers. (**A**) Three dimensional image, viewed dorsally, of the depressor tendons and proximal hind leg joints of a nymph of *Stenocranus acutus*. The helical end of the tendon on which the main body of fibres of the trochanteral depressor muscle (M133b/c) insert, protrudes in the thorax beyond the anterior edge of the coxa. The trochantera and femora of both hind legs are in their

*Figure 6 continued on next page*

*Figure 6 continued*
almost fully levated position so that the external coxal protrusions are close to engaging with the femora. The trochantera were displaced laterally so that their cog wheels are not engaged. (B) The tendon and proximal joints of the right side of the thorax of a nymph of *Metcalfa pruinosa* viewed medially. (C) The tendon and proximal joints of the right side of the thorax of a nymph of *Acanalonia conica* viewed ventrally. (D) The ventral wall of the right coxa of an adult *Metcalfa pruinosa* was removed to reveal the tendon and its insertion on the trochanter. Insertion sites of a small muscle along the straight part of the tendon are visible.
The following figure supplement is available for figure 6:

**Figure supplement 1.** The tendon of the trochanteral depressor muscle in planthoppers.

*Lyons et al., 2011*). We find here that resilin also stains with acridine orange and has a distribution which matches that revealed by all the preceding methods including wide field microscopy with DAPI filters (*Figures 3* and *9* and *Figure 3—figure supplement 1*).

In planthoppers and froghoppers the resilin forms an elongated region in each pleural arch sandwiched between two plates of cuticle with folds and grooves for increased strength. In froghoppers there is a second region of resilin and where it overlaps with the first, a pleural arch in sections appears like a double-decker sandwich. The two fillings of resilin are separated from each other by a middle layer and bounded by two outside layers of cuticle. Relative to body mass, the volume of resilin in froghoppers is 50% higher than in planthoppers. The large regions of resilin are of comparable volume in the two groups so that the increase in volume in froghoppers is explained by the presence of the second region. In its best jumps an adult male planthopper, *Issus coleoptratus* (*Burrows, 2006a*, *2009*), that is 22% heavier than a froghopper, *Lepyronia quadrangularis*, expends 55% more energy and requires a power output that is 213% larger to achieve a take-off velocity that is 20% higher. Why then do froghoppers have more resilin when their energy requirements for jumping are less? The answer may lie in the fact that the composite material from which the energy stores are made endows the jumping mechanism with three key features. First, the pleural arches do not fracture when bent by the considerable force generated by the powerful muscle contractions. Second, the elasticity of the resilin enables the original body shape to be restored rapidly after each jump, so that subsequent jumps can performed quickly. Third, the same muscle contractions will produce the same distortion of the energy stores and hence enable predictable jumping performances. The different volume and distribution of resilin in the two groups may thus reflect possible differences in the distribution of bending forces within their complex pleural arches.

## Muscles, tendons and the storage and release of energy

The three dimensional images of the depressor tendon of planthoppers reveal dramatic changes in its mechanical advantage relative to the joint between the trochanter and the coxa as it moves from its most levated position before a jump to its most depressed position after propelling the jump. As the joint is depressed, the depressor lever arm becomes larger thus increasing its mechanical advantage over the smaller levator muscle. Likewise, as the joint is levated the lever arm of the depressor becomes smaller but does not change sign. Thus when fully levated and cocked in advance of a jump the action of the depressor muscle would still be to cause depression. The same conclusion based on physiological experiments has been drawn for the actions of the depressor muscle in froghoppers (*Burrows, 2006b*). These observations thus do not support the model that has been proposed for jumping in froghoppers (*Gorb, 2004a2004*). In Gorb's model the initial action of the depressor muscle is proposed to levate the trochanter into its fully cocked position before the jump. The sign of the lever arm is then changed by a trigger muscle called M5 (possibly the same as small depressor muscle M133a described here) so that the action of the main depressor is now to depress the hind leg in its movements that propel a jump. In other words the tendon is initially on one side of the axis of rotation of the joint so that it causes levation and the contraction of the trigger muscle then pulls it to the other side so that it now causes depression and propulsion of the jump. Similar kinds of triggering mechanisms have been proposed for jumping in fleas (*Bennet-Clark and Lucey, 1967*), the strike of mantis shrimps (*Cox et al., 2014a2014*) and the snapping movements of alpheid shrimps (*Ritzmann, 1974a1974*), but no definitive evidence has been provided for lever arms

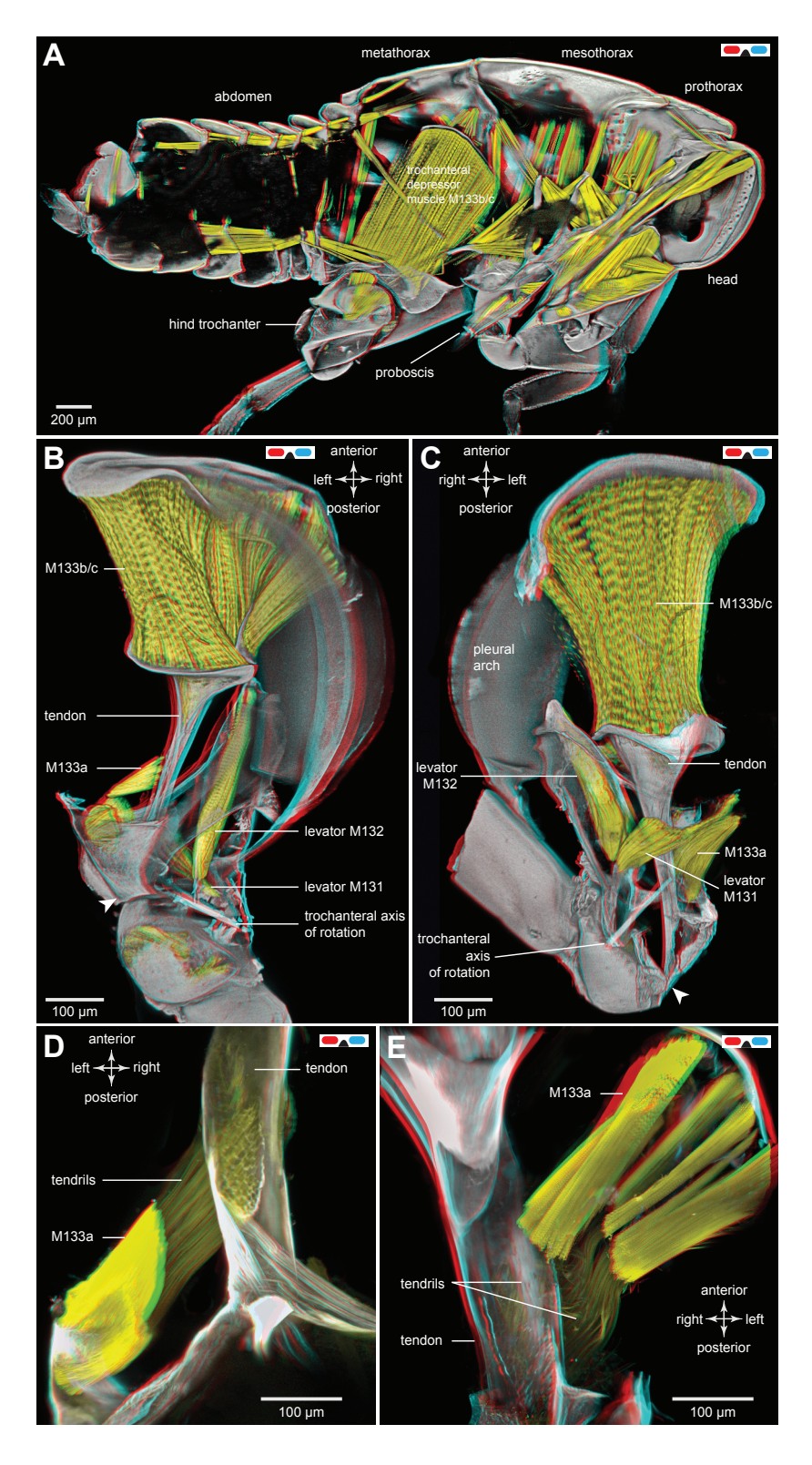

**Figure 7.** Muscles controlling movements of the hind trochantera of planthoppers. (**A**) Medial view of a whole nymph of *Metcalfa pruinosa*. The muscles are shown in yellow. (**B**) Dorsal view of the right metathorax of a nymph with the right hind leg almost fully depressed, showing: the large trochanteral depressor muscle (M133b/c) with its origin in the anterior metathorax, the smaller part of this muscle (M133a) originating on the medial wall of the coxa, and a trochanteral levator muscle (M132). The two parts of the depressor muscle insert on different regions of the same tendon. The white arrowhead

*Figure 7 continued on next page*

*Figure 7 continued*

here (and in C) shows the insertion of the tendon on the medial wall of the trochanter. The line between the dorsal and ventral pivots of the trochanter indicates the axis of its rotation about the coxa. (C) The right half of the thorax viewed ventrally and with the hind leg fully levated. The axis of trochanteral rotation is again indicated by the line between the pivots. (D) Dorsal view of the insertions, by narrower tendrils, of fibres from the small part of the muscle (M133a) onto the main tendon in an adult. The hind leg was depressed as in (B). (E) Ventral view of these same insertions.

The following figure supplement is available for figure 7:

**Figure supplement 1.** Muscles controlling movements of the hind trochantera of planthoppers.

changing in sign. Were such a trigger mechanism to exist in planthoppers and froghoppers it would require close neural synchronisation between the muscles of both hind legs to ensure that an uncontrolled spin of the body does not result (*Sutton and Burrows, 2010a*). Evidence from both adult (*Burrows, 2010*) and larval (*Burrows and Sutton, 2013*) planthoppers suggests the accuracy of the synchrony that would be required can only be achieved mechanically, but no evidence for such a mechanism has been revealed here that could trigger the release of the two hind legs.

Our results suggest a different mechanism. In preparation for a jump the hind legs are levated by the contractions of the levator muscles, an action confirmed by recordings from these muscles in both planthoppers and froghoppers (*Burrows, 2007*; *Burrows and Bräunig, 2010*). At the fully levated or cocked position of the joint, the small levator muscle has a big mechanical advantage and could thus restrain the large depressor which now has its smallest mechanical advantage. When the levator muscle starts to relax, the joint will begin to depress and with further movement the mechanical advantage of the levator will decrease while that of the depressor increases. The changes in mechanical advantages will quickly reach a point where the energy stored by the depressor will result in a rapid depression of the leg to propel a jump. This suggests that in natural jumps by planthoppers there is no need to invoke the action of a trigger muscle. This indicates that the functional role of the small part (M133a) of the depressor muscle is not to effect a change in the sign of the lever arms, but possibly to stabilise the tendon while it delivers the enormous force for jumping.

Support for the release of stored energy in this way also comes from experiments on other insects and from modelling studies. In locusts, jumping is propelled by a large power producing extensor muscle that operates with a smaller flexor muscle and with energy stored in distortions of specialised cuticular structures (*Bennet-Clark, 1975a1975*; *Heitler, 1974*) also made of composites of resilin and hard cuticle (*Burrows and Sutton, 2012*). When flexed in preparation for a jump, the contraction of the small flexor can restrain the joint from moving while both it and the larger extensor co-contract to store energy. When the flexor relaxes the tibia starts to extend and the increasing mechanical advantage of the extensor is expressed in rapid movements of the tibia. Theoretical modelling has also suggested a similar mechanism for jumping in bullfrogs that also involves elastic elements and increasing mechanical advantage during the muscle contractions (*Roberts and Marsh, 2003*), and more generally for muscular systems with these features (*Galantis and Woledge, 2003a2003*).

Insights into the functional mechanisms underlying jumping have been gained by revealing the three dimensional structure of the energy storage devices, muscles, and tendons. These emphasise the important interplay between

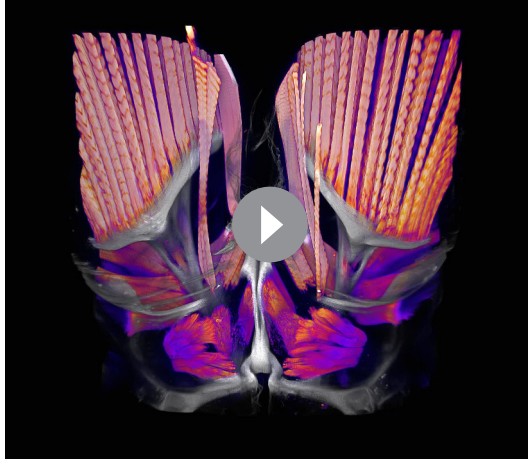

**Video 3.** Three dimensional rotational view of the jumping muscles in an adult planthopper *Stenocranus acutus*.    Both hind legs are in the fully levated position. The small parts of the trochanteral depressor muscles (M133a and M133d) are best viewed during the ventral portion of the rotation and the levator muscles (M131, M132) during the dorsal part.

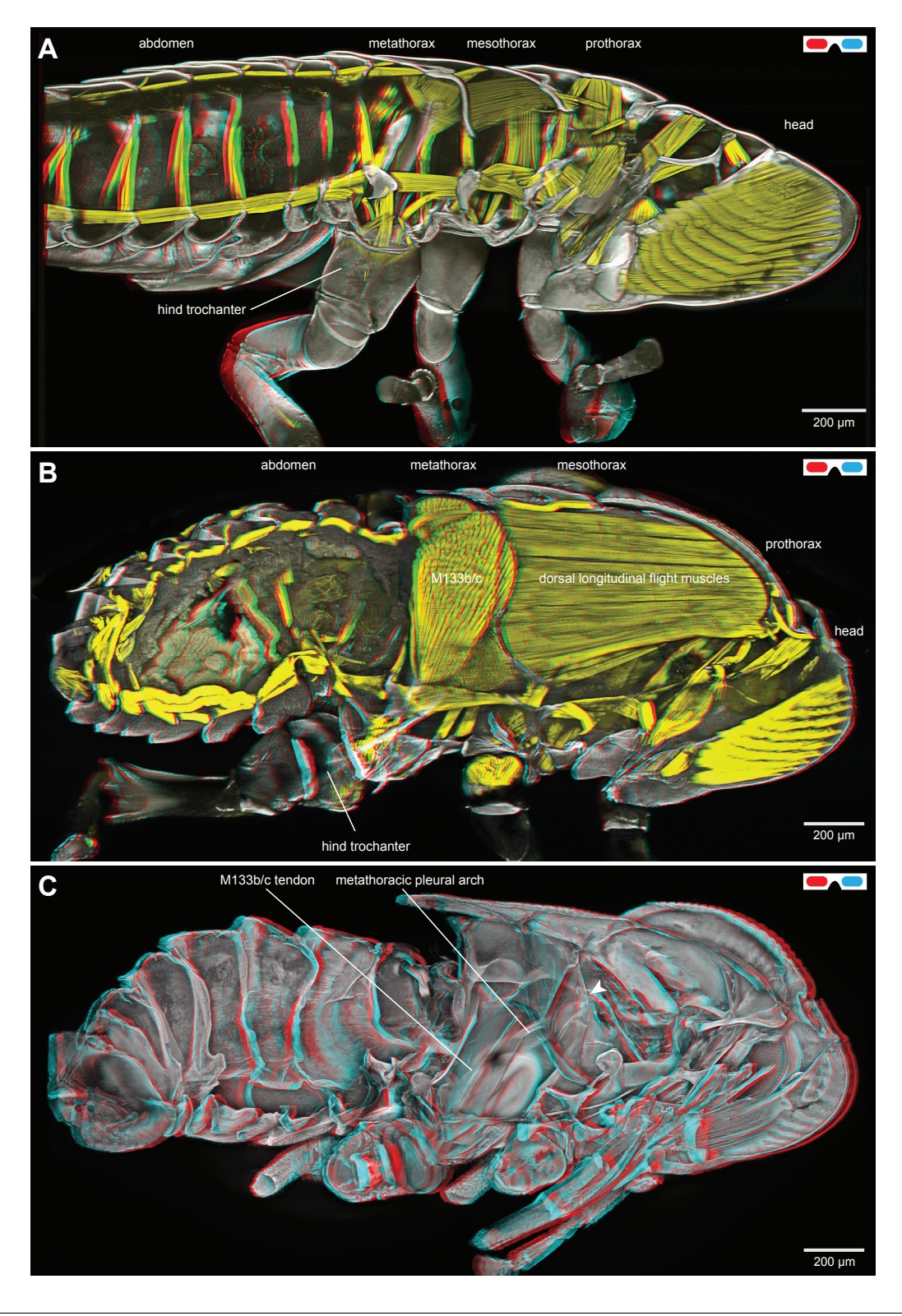

**Figure 8.** Thoracic muscles in nymphal and adult froghoppers. (**A**) Medial view of a nymph of the froghopper *Clastoptera obtusa*. (**B**) Medial view of an adult of this species showing the large metathoracic trochanteral depressor muscle, and large mesothoracic flight muscles. (**C**) The same specimen as in (**B**) to show the internal skeleton of the thorax after removal of the soft tissues. The articulation of the metathoracic pleural arch with the mesothoracic pleural arch at the origin of the trochanteral depressor M133c is marked with a white arrowhead.

*Figure 8 continued on next page*

*Figure 8 continued*

The following figure supplement is available for figure 8:

**Figure supplement 1.** Thoracic muscles in nymphal and adult froghoppers.

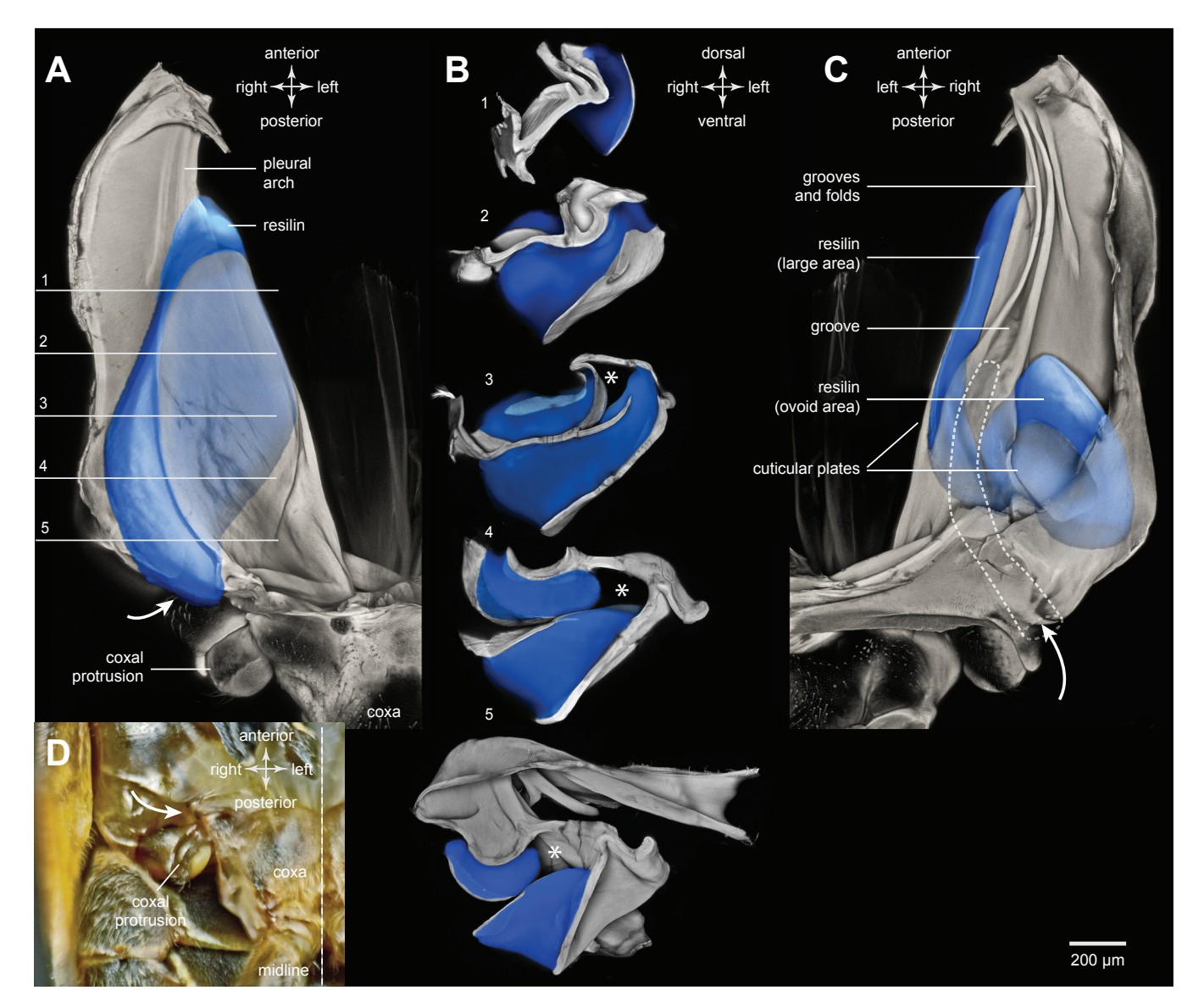

**Figure 9.** Resilin and the pleural arches of an adult froghopper. (A) Ventral view of the right pleural arch of *Lepyronia quadrangularis*. The large region of resilin (blue) is sandwiched between an inner and an outer layer of cuticle. (B) Sections cut at the five levels indicated by the lines and numbers in (A). The asterisks indicate a tunnel-like space in the pleural arch. (C) Dorsal view of the right pleural arch in which two regions of resilin are visible; the first, long and medial strip is the edge of the large region visible ventrally in (A); the second, ovoid region is more lateral and posterior, and is separated from the first by a thin central layer of cuticle and is bounded by an outer layer of cuticle. The dashed outline indicates the path of the tunnel from its entrance near the coxal protrusion (curved white arrow) to its blind ending at the level where the groove forms in the medial plate of the pleural arch. (D) Photograph of the ventral surface of the right side of the metathorax to show the external entrance to the tunnel (curved white arrow) relative to the coxal protrusion and the right hind coxa.

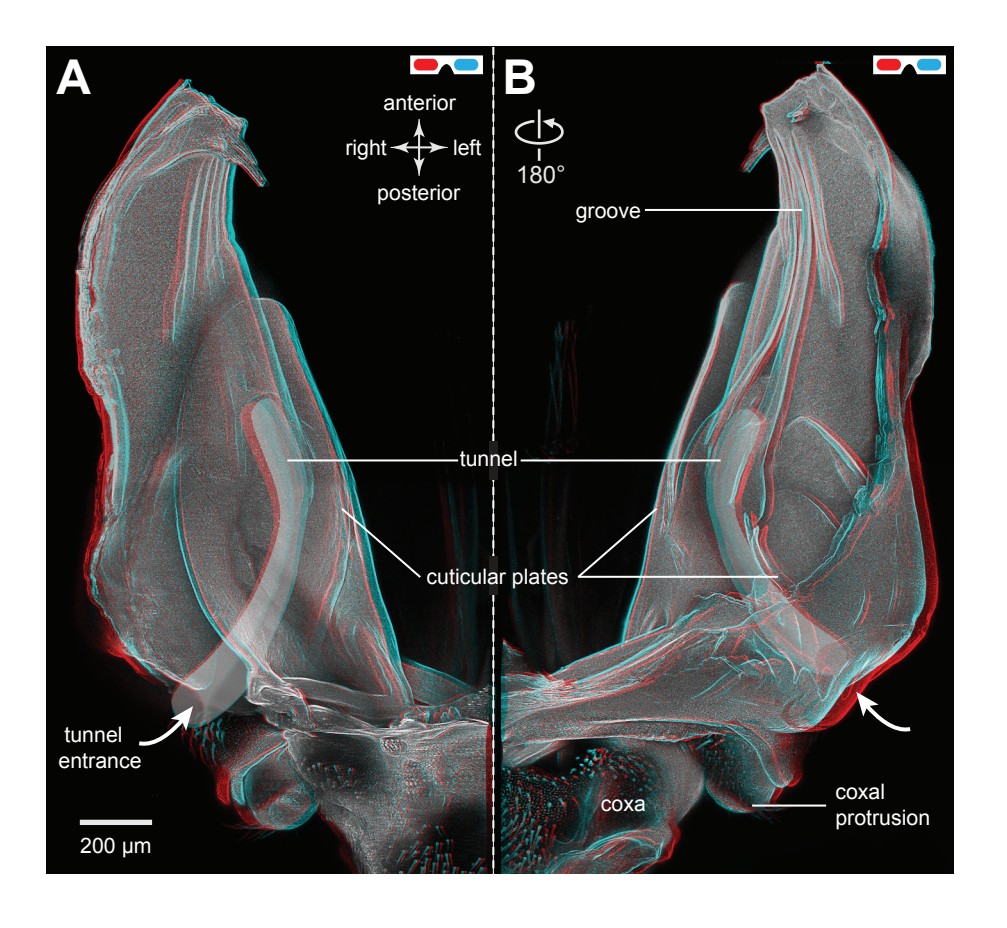

**Figure 10.** Cuticular plates of a pleural arch that surround the resilin in an adult froghopper. (**A**) Three dimensional ventral view of a right pleural arch of *Lepyronia quadrangularis* showing the medial and lateral plates of cuticle that normally encase the large region of resilin. (**B**) Three dimensional dorsal view of the right pleural arch showing the more posterior plates that normally encase the smaller ovoid region of resilin. A drawing of the tunnel that connects with the outside air (curved white arrows) near the coxal protrusion is superimposed on both (**A**) and (**B**). The structures were imaged using red autofluorescence.

The following figure supplement is available for figure 10:

**Figure supplement 1.** Cuticular plates of a pleural arch that surround the resilin in an adult froghopper.

brain and body in producing an adaptive movement. The storage of energy for jumping is only possible because of the mechanics of the changing lever arms of the controlling muscles, the composite materials of the skeletal energy stores and the actions of the muscles that are controlled by the nervous system. These powerful insights suggest that the same methods may be of wider application in the study of complex yet miniaturised mechanisms in small animals.

## Materials and methods

### Insects

The Hemipteran insects, suborder Auchenorrhyncha, studied in this paper were collected from similar habitats containing a variety of grasses, wild flowers and shrubs around Ashburn, Virginia, USA and Cambridge, UK in 2014- 2016. They belong to two groups: first, the Fulgoroidea, the planthoppers, of which there are some 20 families; members of six families which are all adept jumpers with

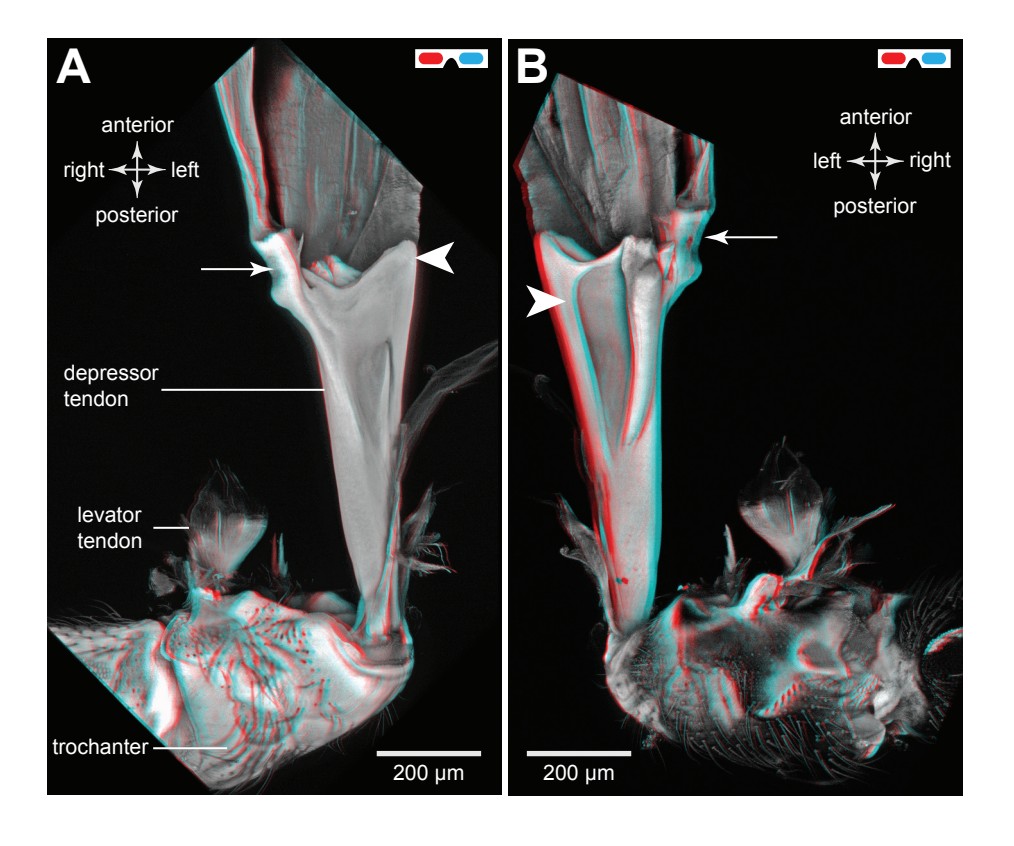

**Figure 11.** The tendon of the trochanteral depressor muscle in an adult froghopper. (**A**) Three dimensional ventral view of the tendon of *Lepyronia quadrangularis* as it passes from the thorax, through the coxa to insert on the medial wall of the trochanter. Within the thorax the tendon bifurcates into a thick branch (arrowhead) onto which the fibres of the depressor muscle (M133c) inserts and a thinner lateral branch (arrow) on which the pinnate fibres of M133b insert. (**B**) Three dimensional dorsal view of the same preparation as in (**A**). Shorter tendons of the trochanteral levator muscles are also shown. The midline is close to the right of the image in (**A**) and to the left of the image in (**B**).

The following figure supplement is available for figure 11:

**Figure supplement 1.** The tendon of the trochanteral depressor muscle in an adult froghopper.

similar thoracic anatomies were studied here: 1. Acanaloniidae, *Acanalonia conica* (Say 1830). 2. Caliscelidae, *Bruchomorpha oculata* (Newman 1838). 3. Delphacidae, *Stenocranus acutus* (Beamer 1946), and *Pissonotus* sp. 4. Derbidae, *Apache degeeri* (Kirby 1821). 5. Flatidae, *Metcalfa pruinosa* (Say 1830) and *Flatormenis proxima* (Walker 1851). 6. Issidae, *Thionia bullata* (Say 1830), and *Issus coleoptratus* (Fabricius 1781). Second, the Cercopoidea, (froghoppers) of which there are three families. The froghopper species used here were from the Cercopidae, *Lepyronia quadrangularis* (Say 1825) and Clastopteridae, *Clastoptera obtusa* (Say 1825). Both are adept jumpers with similar thoracic anatomies.

## Sample preparation

For the imaging of cuticle, an insect was anaesthetized on ice and its thorax was isolated in phosphate buffered saline (PBS). The muscles were then digested away with a mixture of hyaluronidase/trypsin at 0.5 mg/ml for 5 hr at 37°C. Resilin of planthoppers and froghoppers was resistant to trypsin, but not to protease K or collagenase. To speed digestion and produce clean cuticle an ultrasonic bath was used towards the end of the process. To render thick samples transparent for imaging it

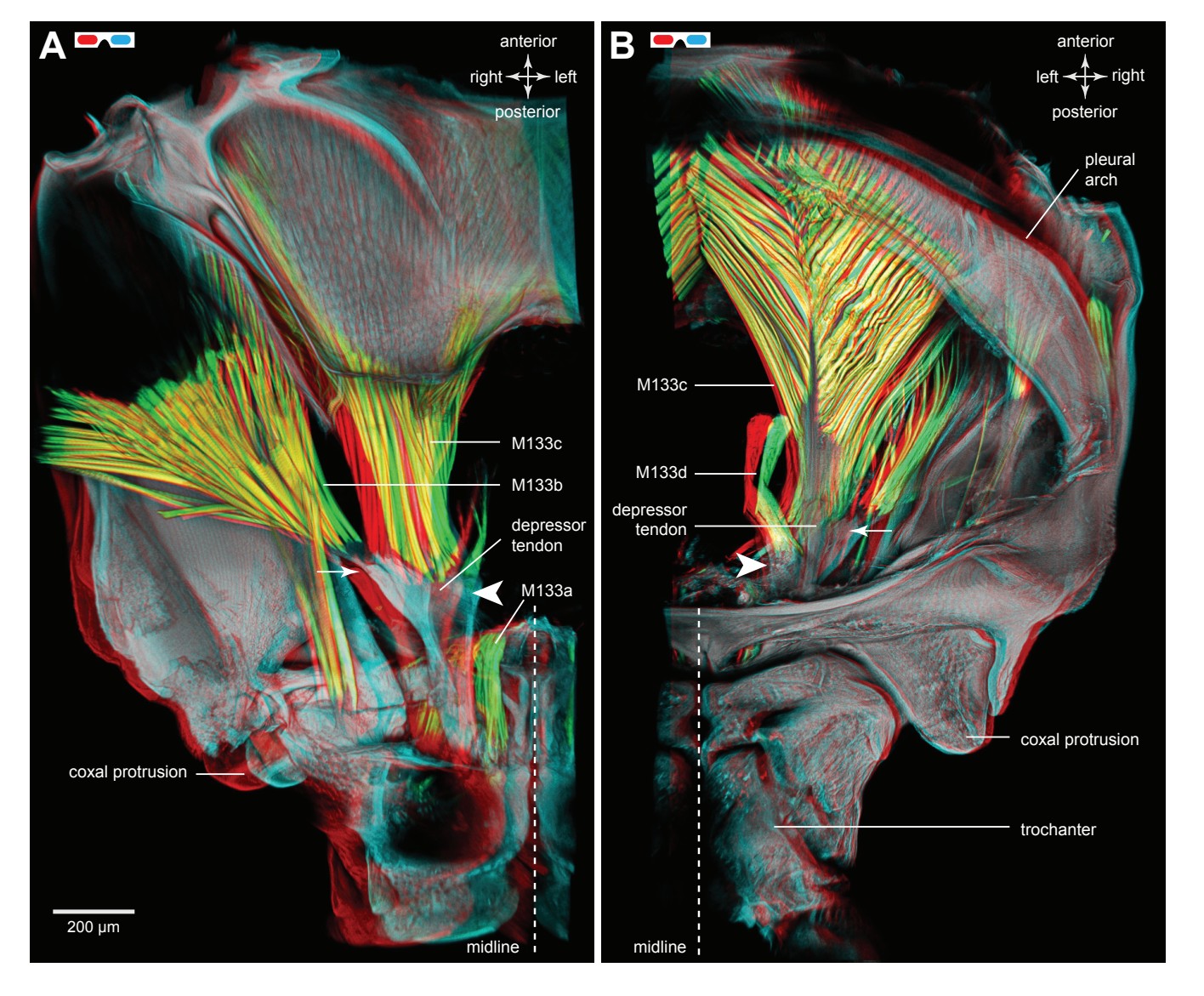

**Figure 12.** Muscles controlling movement of the hind trochantera of adult froghoppers. (**A**) Ventral view of the right side of the metathorax of *Clastoptera obtusa*. The insertion site of muscle M133b is occluded by darkly-pigmented cuticle. (**B**) Dorsal view of the right side showing the insertion of M133d and the pinnate arrangement of fibres in M133b.

The following figure supplement is available for figure 12:

**Figure supplement 1.** Muscles controlling movement of the hind trochantera of adult froghoppers.

was necessary to match the refractive index of the tissue with that of the mounting medium. The specimens were washed with ethanol and mounted in methyl salicylate (Sigma-Aldrich, M6752). Energy stores prepared in this way were also embedded in 8% agarose and sectioned on Leica Vibratome (VT1000s) at 0.3 mm using the slowest settings for progression of the blade. The sections were mounted in Tris-buffered (50 mM, pH 8.4) 50% glycerol.

To image cuticle and muscles, the thorax was fixed in 2% paraformaldehyde/PBS for 24–48 hr at room temperature. After washing in PBS containing 1% triton X-100, samples were treated with 0.5 mg/ml hyaluronidase for 2–3 hr at 37°C and briefly sonicated in an ultrasound bath to remove other tissue – mostly fat bodies – in the body cavity. This tissue caused much light scattering so that its

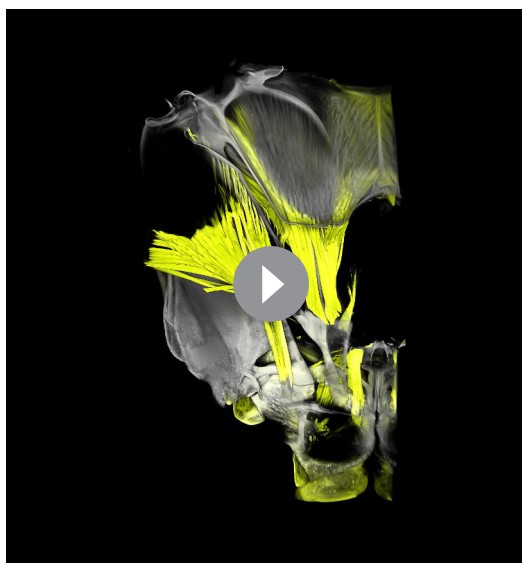

**Video 4.** Three dimensional rotational view of the jumping muscles in an adult froghopper *Clastoptera obtusa*.        Right half of the metathorax with the leg fully extended. The origins of depressor muscle (M133b) are obscured by dark, opaque cuticle.

removal was essential to obtain clear images of muscles and cuticle. The samples were then stained with Texas Red-X Phalloidin (1:50, Life Technologies #T7471) in PBS with 1% triton X-100, 0.5% DMSO and Escin (0.05 mg/ml, Sigma-Aldrich, E1378) at room temperature with agitation for 4–5 days. Long incubation times and the presence of surfactants assured better penetration of phalloidin into the large muscles. In preparations where the 405 nm laser-induced autofuorescence of exoskeletal elements, such as fine tendons in nymphs, was not strong, a chitin-binding dye Calcofluor White (0.1 mg/ml, Sigma-Aldrich, F3543-1G) was included in the staining procedure. Acridine orange and Calcofluor White were never used in the same preparation. The samples were then washed in PBS/1% triton and fixed for 4 hr in 2% paraformaldehyde to reduce leaching of bound phalloidin from muscles during the subsequent ethanol dehydration step. To avoid artefacts caused by osmotic shrinkage of soft tissue, samples were gradually dehydrated in glycerol (2–80%) and then ethanol (20% to 100%) (*Ott, 2008*) and mounted in methyl salicylate for imaging.

We attempted to apply the imaging regime/method for visualization of resilin in the exoskeleton of insects as described by Michels and Gorb (*Michels and Gorb, 2012*) but failed to obtain satisfactory separation of the resilin and chitin autofluorescence in adult plant- and froghopper energy stores (*Figures 3* and *9*). The lack of satisfactory contrast necessitated the use of staining. To reveal the distribution of the elastic protein resilin, samples were stained overnight with Acridine Orange (0.01 mg/ml, Sigma, A-6014) in PBS with 1% triton X-100. The distribution of this staining was then compared with the characteristic blue fluorescence of resilin seen under an Olympus MVX10 Microscope equipped with DP70 color camera and DAPI filter set (excitation: 352–402 nm, emission: 414–477 nm) (*Figure 3—figure supplement 1*) (*Burrows et al., 2011*, *2008*). The distribution of resilin revealed by both methods were the same.

## Imaging and rendering

Serial optical sections were obtained at 2 μm intervals on a Zeiss 710 confocal microscope with a Plan-Apochromat 10x/0.45 NA objective, or at 1 μm with a LD-LCI 25x/0.8 NA objective For the imaging of Acridine Orange-stained cuticle, laser lines of 405 and 488 nm were used. Autofluorescence of chitin was recorded in the 410–465 nm range whereas fluorescence of Acridine Orange was recorded in the 520–580 nm range. The dye exhibits a more red-shifted fluorescence compared to chitin so that collecting emitted light in the 520–580 nm range reduced the chitin autofluorescence component in the registered images. To reduce further the contribution of chitin autofluorescence, we adjusted the offset during recording. Calcofluor White and Texas Red phalloidin-treated samples were imaged using a 405 and 594 nm lasers, respectively. Using these wavelengths for visualization of Calcofluor white-stained exoskeleton and Acridine Orange (AO) -stained resilin creates the possibility of crosstalk. The autofluorescence of resilin would have been included in the visualization; we avoided that by applying Calcofluor White staining only to the nymphs that lack resilin in their energy stores. The settings used to image AO-resilin are appropriate for visualization of exoskeleton structures with large proportions of chitin (*Michels and Gorb, 2012*). The intensity of AO fluorescence is much stronger than autofluorescence of chitin, however, so that any crosstalk did not play an important role in our analysis of the distribution of resilin and did not impact the results.

We did not notice substantial differences in the distribution of blue, green and red autofluorescence in cuticular samples. Red autofluorescence was preferentially used for visualization of chitinous

elements of the energy stores alone (i.e. when muscles and resilin were not simultaneously imaged), as longer wavelengths are less affected by cuticle pigmentation and thickness, yielding more uniform signal throughout the sample. Autofluorescence of cuticle was excited with a 561 nm laser line and recorded in the 568–683 nm range. The stains and excitation and emission wavelengths used for imaging specimens are listed in *Table 1*. For the 10x air objective, the Z-spacing was corrected for the refractive index of the mounting medium (*Ott, 2008*). Images were processed in Fiji (http://fiji.sc/), Icy (http://icy.bioimageanalysis.org/) and Photoshop (Adobe Systems Inc.). Red/cyan anaglyphs (three dimensional representations) of the confocal stacks were created using the stereo pair plug-in of ImageJ. The 3-D images should be viewed with glasses in which the left eye has a red filter and the right eye a cyan filter. Images which should be viewed in this way are indicated in the Figures by the symbol of a pair of coloured glasses. 2-D versions of these Figures are also given. In the renderings of confocal images we use blue to indicate resilin (reflecting its characteristic blue fluorescence in UV light), yellow for muscles (it is the only colour showing good contrast in red-cyan anaglyphs) and shades of grey for cuticle.

The metathoracic muscles involved in jumping are given the same names and numbers (e.g. trochanteral depressor muscle M133 a-d, indicating four different parts of the same muscle) as those used for locusts (*Snodgrass, 1929*, *Snodgrass, 1935*), because of the strong similarities found between these insects (*Bräunig and Burrows, 2008*).

Sequential images of jumping movements of *Issus coleoptratus* were captured by a Photron Fastcam SA3 high speed camera (Photron (Europe), Marlow, Buckinghamshire, UK) at rates of 30 000 frames s$^{-1}$ and with an exposure time of 0.03 ms. The camera was mounted on a Zeiss Stemi SV6 stereo microscope.

## Acknowledgements

We thank Greg Sutton for help in calculating the volumes of resilin and advice on lever arms, and Cambridge and Janelia colleagues for many helpful suggestions during the experimental work and for their constructive comments on drafts of the manuscript. This work was supported by the Howard Hughes Medical Institute.

## Additional information

### Funding

| Funder | Author |
| --- | --- |
| Howard Hughes Medical Institute | Igor Siwanowicz |

The funders had no role in study design, data collection and interpretation, or the decision to submit the work for publication.

### Author contributions

IS, Conceptualization, Formal analysis, Investigation, Methodology, Writing—original draft, Writing—review and editing; MB, Conceptualization, Formal analysis, Supervision, Investigation, Writing—original draft, Writing—review and editing

### Author ORCIDs

Igor Siwanowicz, http://orcid.org/0000-0001-5819-1530

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
