## [Decision Letter]

Thank you for submitting your article "3-D reconstruction of energy stores for jumping in planthoppers and froghoppers from confocal scanning microscopy" for consideration by *eLife*. Your article has been reviewed by three peer reviewers, and the evaluation has been overseen by a Reviewing Editor and K VijayRaghavan as the Senior Editor. The reviewers have opted to remain anonymous.

The reviewers have discussed the reviews with one another and the Reviewing Editor has drafted this decision to help you prepare a revised submission.

Summary:

This study on planthoppers and froghoppers presents a new functional understanding of jumping that is mechanically insightful, and new, highly detailed anatomy with fantastically rendered images. Using confocal microscopy, physical manipulation, and staining, the authors provide a mechanistic understanding of the interaction of muscle, tendon, and exoskeletal elements, which together conspire to produce synchronous actuation in one of nature's fastest movements.

Essential revisions:

There are a few key concerns that require revision to the manuscript, summarized below.

1) The manuscript suffers a bit by focusing on both new methods (emphasized in the Abstract) and scientific results. The only new method is the staining of resilin with acridine orange, suggesting that the manuscript would benefit most from a stronger focus on the research results. Reviewer #1 has some nice suggestions on how to accomplish such a re-focusing.

2) Further details are required to understand the effectiveness of the use of acridine orange for identifying resilin and to justify why this was required compared to autofluorescence (see reviewer #2).

3) The more familiar language of levers and mechanical advantage when discussing the findings (see reviewer #3).

Reviewer #1:

This paper provides an interesting and necessary contribution to current work on the jumping of insects, specifically those who use gears. The image quality, detail, and presentation is stunning.

I did find the results difficult to read because the Introduction does not prepare the reader for the topics investigated. Specifically, in the Introduction there are three rough hypotheses which are too vague to actually test. I suggest making predictions here instead. Then use these specific predictions to organize your Results and Discussion. For instance, in the first point you mention the thorax, where power generating muscles are located, must be strong enough to withstand high forces. This suggests you will measure strength, perhaps of the exoskeleton? Instead, a prediction related to the anatomical structure of the thorax would be more appropriate based on your results.

It also needs to be clearer why you looked at adults and nymphs and what their differences are. Furthermore, what are the expected differences between planthoppers and froghoppers? Why use multiple species within each? Where are the images of these other species? Are there ecological/behavioral differences that make these comparisons interesting/important?

Related to the first two points, certain definitions and details are left out of the Introduction, like resilin, or the ecology of froghoppers, and are included in later sections. Likewise, information on jumping from Burrows would have been useful in the Introduction and would perhaps help formulate predictions (i.e., planthoppers are heavier than froghoppers, expend more energy, etc.).

In the Abstract you mention 3D printing of the components, but don't mention this in the manuscript. You are correct that your work could be used for this, but it is not clear how. This paper reads a bit like a methods paper (i.e., lacks clear biological predictions), but it has too much detail with regards to the subject (e.g., insect anatomy and structure/function relationships) to be a clear methods paper. Given the data collected, I would prefer this work be focused on structure/function relationships focused around clear predictions.

Finally, although the 3D image rendering is impressive, not every reader has the ability to view these with 3D glasses. Can you make 2D images available, perhaps in the supporting information files?

Reviewer #2:

This manuscript describes the structures involved in the energy storage for jumping of planthoppers and froghoppers in great detail. The 3D reconstructions based on confocal laser scanning microscopy (CLSM) analyses are impressive and provide in fact, as stated by the authors, 'unprecedented detail'. The manuscript is well written and clearly comprehensible, and it adds significant information to the already existing knowledge of the respective structures published in earlier work of Professor Burrows. However, the following aspects and comments have to be considered in the review process:

1) The authors stained the exoskeleton structures with large proportions of resilin using the fluorescence dye Acridine Orange. This dye is well-known and typically applied to stain nucleic acids. The finding that Acridine Orange also stains the protein resilin is surprising and very interesting. Unfortunately, the authors only state that 'The distribution of this staining was then confirmed to be the same as that revealed by the characteristic blue fluorescence of resilin when examined with a sharp-edged (1% transmission limits) excitation band of ultraviolet (UV) light.…' but they do not show this. It is absolutely essential that they clearly demonstrate that resilin can be specifically stained with Acridine Orange. For this purpose, the authors definitely have to show the results of precise analyses revealing that the typical resilin autofluorescence and the fluorescence of Acridine Orange are absolutely colocalised in test samples.

2) The authors have to convincingly explain why they decided to fluorescently stain resilin with Acridine Orange. This seems to be not very appropriate because the excitation and emission peaks of the Acridine Orange fluorescence are in the ranges of those of an autofluorescence that is typically exhibited by arthropod exoskeleton structures with large proportions of chitin. Accordingly, it is very likely that because of crosstalk the exclusive visualization of the Acridine Orange fluorescence and thereby the specific visualization of the structures with large proportions of resilin within exoskeleton samples (which is necessary if exoskeleton structures with large proportions of resilin and exoskeleton structures with large proportions of chitin have to be visualized and 3D reconstructed separately as described in the manuscript) are only possible when a precise emission fingerprinting with detailed λ scans and linear unmixing is performed. Why did the authors not use the resilin autofluorescence? In a relatively large number of earlier studies this autofluorescence has been efficiently used to visualize exoskeleton structures with large proportions of resilin, and it was even demonstrated that a combination of CLSM and an excitation wavelength of 405 nm can be successfully applied to reliably visualize exoskeleton structures with large proportions of resilin (Michels and Gorb, 2012).

3) The crosstalk problem mentioned above is also present (and even particularly pronounced) in the case of Fluorescent Brightener 28 (synonym: Calcofluor White M2R) that was applied by the authors to stain chitin in the exoskeleton and to separately visualize the exoskeleton structures with large proportions of chitin. The excitation and emission properties of the fluorescence of Fluorescent Brightener 28 are comparable to those of the autofluorescence of resilin. Accordingly, it is not easy to separate the different fluorescence signals obtained from Fluorescent Brightener 28 and resilin from each other. Such a separation requires a very precise spectral detection system and a rather precise emission fingerprinting with very detailed λ scans and linear unmixing. In case the particular ZEISS LSM 710 applied by the authors for their study is equipped with a 34-channel QUASAR detection unit, the authors had the opportunity to perform such an emission fingerprinting. Unfortunately, in the manuscript they do not explain at all how the fluorescences were detected. (Which emission wavelength ranges were detected?) Why did the authors decide to apply Fluorescent Brightener 28? They could have used an autofluorescence to visualize the exoskeleton structures with large proportions of chitin. For example, many of such exoskeleton structures exhibit a relatively intense autofluorescence when they are exposed to the light of a 561 nm laser. (Many ZEISS LSM 710 systems are equipped with such a laser.) In this context, another question arises: in the Materials and methods section, the authors mention 'Autofluorescence of cuticle was excited with a 562 nm laser line.' (This must be a mistake because such a laser line is definitely not offered by Carl Zeiss Microscopy. I assume that the authors used a 561 nm laser.), however, this autofluorescence is not mentioned anywhere else in the manuscript. What has happened to the results of this autofluorescence visualization?

In conclusion, the authors have to convincingly explain (2) why they did not perform their study using the autofluorescence of resilin and either an autofluorescence of the exoskeleton structures with large proportions of chitin or a fluorescence dye staining chitin and featuring excitation and emission properties that are clearly different from those of the autofluorescence of resilin and, therefore, enable to absolutely reliably visualize exoskeleton structures with large proportions of chitin and exoskeleton structures with large proportions of resilin separately and (1) how they coped with the crosstalk problem.

4) In the Discussion section of the manuscript, the authors should clearly highlight which advances their 'new methodological approach' made possible. For this purpose, they should properly discuss the new insights into the energy storage structures important for jumping of planthoppers and froghoppers with respect to the knowledge of the respective structures obtained in earlier studies of Professor Burrows.

5) In my opinion, the end of the Discussion section of the manuscript is too abrupt. The authors should add a paragraph with concluding remarks.

6) In the Abstract, the authors write 'This new methodological approach can illuminate how miniaturized components interact and function in complex movements of small animals with the potential for 3-D printing of relevant structures.' The potential for 3D printing is not mentioned anywhere else in the manuscript. This topic should be included in the Discussion section of the manuscript.

Reference

Michels J, Gorb SN (2012) Detailed three-dimensional visualization of resilin in the exoskeleton of arthropods using confocal laser scanning microscopy. Journal of Microscopy 245, 1-16

Reviewer #3:

The manuscript entitled "3-D reconstruction of energy stores for jumping in planthoppers and froghoppers from confocal scanning microscopy" describes the detailed structural features that function to store elastic energy and amplify mechanical power in two groups of jumping insects. This work is stunningly beautiful. The level of anatomical detail is indeed unprecedented and set a new standard for future work aiming to derive functional mechanisms from a structural framework. I believe the methods developed here will be of interest to broad audience and likely represents an important contribution to the field. The one potential shortcoming of the paper is that the work is presented in a descriptive way rather than developed hypotheses. In addition, given that the methodology used and developed here is an important aspect of the contribution it may be worth providing the readers with an assessment of the strengths and weaknesses of the approach in the Discussion section. I have also made some stylistic suggestions as well as highlighted a few places where the language could be more precise.

[Editors' note: further revisions were requested prior to acceptance, as described below.]

Thank you for resubmitting your work entitled "Three dimensional reconstruction of energy stores for jumping in planthoppers and froghoppers from confocal laser scanning microscopy" for further consideration at *eLife*. Your revised article has been favorably evaluated by K VijayRaghavan (Senior editor), a Reviewing editor, and three reviewers.

Please find the reviewers' specific comments below. As you can see, the manuscript has been greatly improved, but there are some remaining issues that need to be addressed before acceptance. For reviewer #1's comments, please try to condense the Discussion further if possible, but more importantly, it would be nice to see a brief conclusion to wrap up the paper. Reviewer #2's comments are more technical in nature and require careful consideration. Reviewer #3 is fully satisfied.

Reviewer #1:

This manuscript has greatly improved. I appreciate the level of detail the authors used to incorporate reviewer suggestions. In particular, the Introduction now prepares the readers for work discussed in the following sections. I feel the Discussion section is still a bit long and abruptly ends, however I recommend this manuscript for publication. One small note, in subsection “Distribution of resilin in the energy stored of adult planthoppers”, family names are used, rather than the common names used prior to these sentences. It would be helpful to add the common names here so readers not versed in insect taxonomy can follow.

Reviewer #2:

The authors clearly improved the manuscript. However, there are still a few inconsistencies in the context of the confocal laser scanning microscopy (CLSM) visualizations. These are described below.

1) In the first version of the manuscript, the authors wrote “For the imaging of Calcofluor white and Acridine Orange- stained cuticle, laser lines of 405 and 488 nm were used, respectively.” Based on this description, I assumed that the exoskeleton parts with large proportions of chitin had in all cases been visualized by using Calcofluor White M2R and that the exoskeleton parts with large proportions of resilin had in all cases been visualized by using Acridine Orange. And I was already wondering for which visualization the authors had used the autofluorescence that was mentioned in the sentence (but mentioned nowhere else). In the current version of the manuscript, the description is different. The authors now write “For the imaging of Acridine Orange- stained cuticle, laser lines of 405 and 488 nm were used. Autofluorescence of chitin was recorded in the 410-465 nm range;.…” And Calcofluor White M2R is now mentioned (2) in the context of the visualization of muscles and exoskeleton parts with large proportions of chitin (subsection “Imaging and rendering) and (1) in the lines where the authors write 'In preparations where the 405 nm laser-induced autofuorescence of exoskeletal elements, such as fine tendons in nymphs, was not strong, a chitin-binding dye Calcofluor White (0.1 mg/ml, Σ-Aldrich, F3543-1G) was included in the staining procedure.' These descriptions make the reader think that the exoskeleton parts with large proportions of chitin were visualized mainly by using autofluorescence and only in the case of too low autofluorescence intensities by using Calcofluor White M2R. Which version is correct? In this context, I have the following questions:

a) Why did the authors use an excitation wavelength of 405 nm and why did they detect fluorescence light with wavelengths of 410-465 nm to visualize exoskeleton parts with large proportions of chitin? The application of quite similar settings (405 nm and 420-480 nm) had been demonstrated to result in the visualization of exoskeleton parts with large proportions of resilin (see Michels and Gorb 2012) but not in the visualization of exoskeleton parts with large proportions of chitin. (In a comparable way, the authors used excitation wavelengths of 352-402 nm and detected fluorescence light with wavelengths of 417-477 nm for their wide-field fluorescence microscopy visualization of the resilin autofluorescence (Subsection “Sample preparation”).) Accordingly, with these settings, the authors did very likely not visualize exoskeleton parts with large proportions of chitin.

b) In subsection “Imaging and rendering”, the authors write 'Autofluorescence of cuticle was excited with a 561 nm laser line.' This autofluorescence is also mentioned in the Table 1. However, it is mentioned nowhere else, and the authors do not describe for which visualization this autofluorescence was used.

2) The authors did not really be responsive to my concerns about the large probability of crosstalk having occurred during the CLSM visualizations. They included information about the detected fluorescence wavelengths in the manuscript, and it is now obvious that the applied detection settings had very likely resulted in the occurrence of crosstalk during the CLSM analyses. For example, the Acridine Orange fluorescence, which was used to visualize the exoskeleton structures with large proportions of resilin, was visualized by applying an excitation wavelength of 488 nm and by detecting fluorescence light with wavelengths of 520-580 nm. However, these settings are very appropriate to also visualize an autofluorescence that is typically excited by weakly or non-sclerotised exoskeleton structures with large proportions of chitin (see Michels and Gorb 2012). Accordingly, the authors cannot be sure that they only visualized structures with large proportions of resilin. In the case of the Calcofluor White M2R fluorescence, which was used to visualize the exoskeleton structures with large proportions of chitin, a similar problem exists: this fluorescence was visualized by applying an excitation wavelength of 405 nm and by detecting fluorescence light with wavelengths of 410-570 nm. With these settings, the autofluorescence of resilin was included in the visualization (see above), and the authors cannot be sure that they only visualized structures with large proportions of chitin.

It is now of course not possible to get rid of this 'crosstalk problem' anymore. The fluorescences of fluorescence dyes are often much more intense than autofluorescences. In such situations, the proportions of autofluorescences within the fluorescence signal can be minor, and it can happen that the autofluorescences do not negatively affect the analyses. In my opinion, it seems that this was the case in the present study. The results are convincing, and it seems that crosstalk did not play an important role and have a negative effect. However, the authors should mention crosstalk. They should include a statement saying that with the CLSM settings that they applied crosstalk might have occurred but, because of the relatively high intensities of the dye fluorescences compared to those of the autofluorescences, very likely did not negatively influence the results.

3) The authors should remove the sub-figures Aiii and Biii from the Figure 3—figure supplement 1 because of the inconsistencies regarding the correct visualization of the resilin autofluorescence (see above). Based on the appearance of these sub-figures, I cannot imagine that the respective results are correct.

4) In the subsection “Imaging and rendering”, the authors should not use the term 'contamination'.

Reference

Michels J, Gorb SN (2012) Detailed three-dimensional visualization of resilin in the exoskeleton of arthropods using confocal laser scanning microscopy. Journal of Microscopy 245, 1-16

Reviewer #3:

The main shortcoming of the original submission was that the text was rather dry and written in a mostly descriptive context based on the visualized morphology. It is worth noting the morphology is some of the most visually stunning, cutting edge, high-resolution imaging I've seen at this scale and therefore is a substantial contribution on its own. However, the paper is now much improved. In particular the new sections in the Discussion elevate the paper beyond descriptive morphology and begin to contextualized the anatomical features in a more mechanical and functional framework. The authors have addressed all of my concerns. I am confident that this paper is a substantial and important contribution.

[Editors' note: further revisions were requested prior to acceptance, as described below.]

Thank you for resubmitting your work entitled "Three dimensional reconstruction of energy stores for jumping in planthoppers and froghoppers from confocal laser scanning microscopy" for further consideration at *eLife*. Your revised article has been favorably evaluated by K VijayRaghavan (Senior editor) and a Reviewing editor.

The manuscript has been improved sufficiently, but there is one remaining issue that needs to be addressed before acceptance, as described below:

Perhaps there was some confusion about reviewer 1's suggestion to provide 2D renderings of 3D images. In your February submission, you suggested that you could provide the images as supplementary material. (Note that this wasn't posed as a question.) Our assumption was that this would be done. Although it is true that 3D glasses are cheap and readily available online, the reality is that some (likely many) readers will not have them on hand, and perhaps won't take the time to order them for a single usage. Thus, providing the images will increase the accessibility of the article. Appendices are allowed by the journal, so it should be straightforward to add an Appendix that includes these images.

In particular, please provide 2D versions of each 3D image that hasn't already been provided in the main text. For example, Figure 1 includes both 3D and 2D versions, but Figure 2 does not. So, Figure 2 should be provided as two-dimensional versions in the Appendix. Please amend the figure legends accordingly, to indicate that 2D versions can be found in the Appendix. Correspondingly, please indicate in the Appendix which figure in the main text the 2D versions refer to.

[Editors' note: further revisions were requested prior to acceptance, as described below.]

Thank you for resubmitting your work entitled "Three dimensional reconstruction of energy stores for jumping in planthoppers and froghoppers from confocal laser scanning microscopy" for further consideration at *eLife*. Your revised article has been favorably evaluated by K VijayRaghavan (Senior editor) and a Reviewing editor.

The manuscript has been improved sufficiently, but there is one remaining issue that needs to be addressed before acceptance, as described below:

The new 2D figures in Appendix 1 look great. (In fact, it must be emphasized that the images in this paper are stunning, worthy of textbooks.) But, all 9 of these figures include 3D glasses icons, which indicates to the reader to use 3D glasses. Was that intended? In the figures in the main body, the glasses icon is only used for 3D-colored images, and not for the 2D images, so the assumption is that this convention holds in the Appendix as well.

---

## [Author Response]

Essential revisions:

There are a few key concerns that require revision to the manuscript, summarized below.

1) The manuscript suffers a bit by focusing on both new methods (emphasized in the Abstract) and scientific results. The only new method is the staining of resilin with acridine orange, suggesting that the manuscript would benefit most from a stronger focus on the research results. Reviewer #1 has some nice suggestions on how to accomplish such a re-focusing.

We have revised the paper so that it is now focused on the scientific results concerning how energy is stored for jumping and have followed the very helpful comments of reviewer 1. We have removed all comments about new techniques. The Introduction now poses three questions about how the storage devices for jumping store and release energy that are then followed through in the same order in Results and the Discussion.

2) Further details are required to understand the effectiveness of the use of acridine orange for identifying resilin and to justify why this was required compared to autofluorescence (see reviewer #2).

We have included a supplementary figure that directly addresses this issue. It shows a lack of sufficient dominance of blue fluorescence from resilin over chitin autofluorescence in planthopper and froghopper pleural arches when using the imaging regime suggested by the reviewer and described in Michels and Gorb, 2012. The absence of contrast between those structural elements necessitated the use of staining. The figure also shows side-by-side comparison of wide-field fluorescence micrograph of the resilin autofluorescence in planthopper and froghopper energy stores and the results of Acridine Orange staining imaged with CLSM. We hope the figure convincingly demonstrates colocalization of the characteristic autofluorescence signal and Acridine Orange staining.

3) The more familiar language of levers and mechanical advantage when discussing the findings (see reviewer #3).

We have changed the language we have used in describing lever arms and mechanical advantages to that suggested by reviewer 3 throughout the paper. We have consulted a mechanical engineer to ensure that our descriptions are correct, and have checked carefully for consistency.

Reviewer #1:

This paper provides an interesting and necessary contribution to current work on the jumping of insects, specifically those who use gears. The image quality, detail, and presentation is stunning.

I did find the results difficult to read because the Introduction does not prepare the reader for the topics investigated. Specifically, in the Introduction there are three rough hypotheses which are too vague to actually test. I suggest making predictions here instead. Then use these specific predictions to organize your Results and Discussion. For instance, in the first point you mention the thorax, where power generating muscles are located, must be strong enough to withstand high forces. This suggests you will measure strength, perhaps of the exoskeleton? Instead, a prediction related to the anatomical structure of the thorax would be more appropriate based on your results.

It also needs to be clearer why you looked at adults and nymphs and what their differences are. Furthermore, what are the expected differences between planthoppers and froghoppers? Why use multiple species within each? Where are the images of these other species? Are there ecological/behavioral differences that make these comparisons interesting/important?

The biology that we have added to the Introduction now explains that nymphal planthoppers but not nymphal froghoppers are adept at jumping like the adults. We have used different species from different families to demonstrate the generalities of the structures and mechanisms within the two groups. The species illustrated are named in each of the figures. The insects analysed were found in similar habitats (but we can’t assume that this is true for the many species that belong to these two groups) and show broadly similar behaviour – particularly as far as jumping – but detailed knowledge is not available in the literature.

Related to the first two points, certain definitions and details are left out of the Introduction, like resilin, or the ecology of froghoppers, and are included in later sections. Likewise, information on jumping from Burrows would have been useful in the Introduction and would perhaps help formulate predictions (i.e., planthoppers are heavier than froghoppers, expend more energy, etc.).

We have moved all these points to the Introduction and have expanded some to give the reader a broader background to this study.

In the Abstract you mention 3D printing of the components, but don't mention this in the manuscript. You are correct that your work could be used for this, but it is not clear how.

We have deleted this superfluous mention of 3D printing.

This paper reads a bit like a methods paper (i.e., lacks clear biological predictions), but it has too much detail with regards to the subject (e.g., insect anatomy and structure/function relationships) to be a clear methods paper. Given the data collected, I would prefer this work be focused on structure/function relationships focused around clear predictions.

We have refocused the paper so that we hope it no longer reads a bit like a methods paper. It was always our intention that the focus was on answering specific questions of how the energy for jumping was stored. We would hope that readers will be encouraged by the power of the methodology we have used when applied to functional analyses of movement.

Finally, although the 3D image rendering is impressive, not every reader has the ability to view these with 3D glasses. Can you make 2D images available, perhaps in the supporting information files?

We have agonized over this problem (although the glasses cost only a few pennies from Amazon) but did not know how to resolve it. Your suggestion is very helpful and if the journal would allow us to do this, we would be happy to duplicate the 3D figures in 2D for the supplementary material. But there would of course be a loss of some information that only the glasses could bring out.

Reviewer #2:

This manuscript describes the structures involved in the energy storage for jumping of planthoppers and froghoppers in great detail. The 3D reconstructions based on confocal laser scanning microscopy (CLSM) analyses are impressive and provide in fact, as stated by the authors, 'unprecedented detail'. The manuscript is well written and clearly comprehensible, and it adds significant information to the already existing knowledge of the respective structures published in earlier work of Professor Burrows. However, the following aspects and comments have to be considered in the review process:

1) The authors stained the exoskeleton structures with large proportions of resilin using the fluorescence dye Acridine Orange. This dye is well-known and typically applied to stain nucleic acids. The finding that Acridine Orange also stains the protein resilin is surprising and very interesting. Unfortunately, the authors only state that 'The distribution of this staining was then confirmed to be the same as that revealed by the characteristic blue fluorescence of resilin when examined with a sharp-edged (1% transmission limits) excitation band of ultraviolet (UV) light.…' but they do not show this. It is absolutely essential that they clearly demonstrate that resilin can be specifically stained with Acridine Orange. For this purpose, the authors definitely have to show the results of precise analyses revealing that the typical resilin autofluorescence and the fluorescence of Acridine Orange are absolutely colocalised in test samples.

We have included a supplementary figure showing side-by-side comparison of wide-field fluorescence micrograph of the resilin autofluorescence in planthopper and froghopper pleural arches and the results of Acridine Orange staining imaged with CLSM. In this comparison the natural fluorescence of the resilin matches exactly with the staining with AO. Please note that the differences in the distribution of resilin in planthoppers and froghoppers is also visible in both the autofluorescence images and the AO staining. We hope the figure convincingly demonstrate colocalization of the characteristic autofluorescence signal and Acridine Orange staining.

2) The authors have to convincingly explain why they decided to fluorescently stain resilin with Acridine Orange. This seems to be not very appropriate because the excitation and emission peaks of the Acridine Orange fluorescence are in the ranges of those of an autofluorescence that is typically exhibited by arthropod exoskeleton structures with large proportions of chitin. Accordingly, it is very likely that because of crosstalk the exclusive visualization of the Acridine Orange fluorescence and thereby the specific visualization of the structures with large proportions of resilin within exoskeleton samples (which is necessary if exoskeleton structures with large proportions of resilin and exoskeleton structures with large proportions of chitin have to be visualized and 3D reconstructed separately as described in the manuscript) are only possible when a precise emission fingerprinting with detailed λ scans and linear unmixing is performed. Why did the authors not use the resilin autofluorescence? In a relatively large number of earlier studies this autofluorescence has been efficiently used to visualize exoskeleton structures with large proportions of resilin, and it was even demonstrated that a combination of CLSM and an excitation wavelength of 405 nm can be successfully applied to reliably visualize exoskeleton structures with large proportions of resilin (Michels and Gorb, 2012).

We have indeed tried to use the imaging procedure described by Michels and Gorb in their beautiful 2012 work, but we were unable to observe a pronounced dominance of blue fluorescence from resilin. We therefore did not obtain satisfactory contrast between highly sclerotized cuticle and resilin in these energy stores. We included the examples of our attempts in the supplementary figure (panel ii). Instead we found out that resilin stains very strongly with relatively dilute solution of Acridine Orange and the emitted fluorescence is overwhelmingly stronger than that of the surrounding chitinous structures. We now show in Figure 1 in the supplementary material (see above response to point 1) a very strong correspondence between the distribution of resilin as revealed by its autofluorescence and its staining with AO. It is also advantageous that AO exhibits a more red-shifted fluorescence compared to chitin; collecting emitted light in the 520-580 nm range reduced the chitin autofluorescence component in the registered images. To further reduce the contamination of AO-resilin signal from chitin autofluorescence we adjusted the offset during recording. We have added this further information to the Methods section.

3) The crosstalk problem mentioned above is also present (and even particularly pronounced) in the case of Fluorescent Brightener 28 (synonym: Calcofluor White M2R) that was applied by the authors to stain chitin in the exoskeleton and to separately visualize the exoskeleton structures with large proportions of chitin. The excitation and emission properties of the fluorescence of Fluorescent Brightener 28 are comparable to those of the autofluorescence of resilin. Accordingly, it is not easy to separate the different fluorescence signals obtained from Fluorescent Brightener 28 and resilin from each other. Such a separation requires a very precise spectral detection system and a rather precise emission fingerprinting with very detailed λ scans and linear unmixing. In case the particular ZEISS LSM 710 applied by the authors for their study is equipped with a 34-channel QUASAR detection unit, the authors had the opportunity to perform such an emission fingerprinting. Unfortunately, in the manuscript they do not explain at all how the fluorescences were detected. (Which emission wavelength ranges were detected?) Why did the authors decide to apply Fluorescent Brightener 28? They could have used an autofluorescence to visualize the exoskeleton structures with large proportions of chitin. For example, many of such exoskeleton structures exhibit a relatively intense autofluorescence when they are exposed to the light of a 561 nm laser. (Many ZEISS LSM 710 systems are equipped with such a laser.) In this context, another question arises: in the Materials and methods section, the authors mention 'Autofluorescence of cuticle was excited with a 562 nm laser line.' (This must be a mistake because such a laser line is definitely not offered by Carl Zeiss Microscopy. I assume that the authors used a 561 nm laser.), however, this autofluorescence is not mentioned anywhere else in the manuscript. What has happened to the results of this autofluorescence visualization?

In conclusion, the authors have to convincingly explain (2) why they did not perform their study using the autofluorescence of resilin and either an autofluorescence of the exoskeleton structures with large proportions of chitin or a fluorescence dye staining chitin and featuring excitation and emission properties that are clearly different from those of the autofluorescence of resilin and, therefore, enable to absolutely reliably visualize exoskeleton structures with large proportions of chitin and exoskeleton structures with large proportions of resilin separately and (1) how they coped with the crosstalk problem.

We were not able to obtain satisfactory separation of the resilin and chitin autofluorescence in adult plant- and froghopper energy stores. The autofluorescence of their highly sclerotised exoskeleton of the adults was intense enough when either excitatory light of 405, 488 or 561 nm was used; when imaging chitinous elements of the pleural arches alone we did use 561 nm laser line (we have corrected our mislabeling of this wavelength). In contrast to the adults, 405 nm laser-induced autofuorescence of exoskeletal elements such as fine tendons in nymphs was not strong enough and we decided to augment it with Calcofluor. The blue-green fluorescence of Clacofluor does not bleed into Texas Red Phalloidin channel and we successfully used this combination of dyes for simultaneous 3-D imaging of exoskeleton and muscles. We did not combine Acridine Orange and Calcofluor, opting instead for 405 nm- excited autofluorescence of chitin (see above). To make our methodology clear we have added a table in the Materials and methods section, listing the stains and wavelengths used to image different tissues.

4) In the Discussion section of the manuscript, the authors should clearly highlight which advances their 'new methodological approach' made possible. For this purpose, they should properly discuss the new insights into the energy storage structures important for jumping of planthoppers and froghoppers with respect to the knowledge of the respective structures obtained in earlier studies of Professor Burrows.

We have now deleted all suggestions that this is a new methodological approach and instead focus on the answers that we pose in the Introduction and how these relate to the problems of jumping with such speed and power.

5) In my opinion, the end of the Discussion section of the manuscript is too abrupt. The authors should add a paragraph with concluding remarks.

We have rewritten the last section of the Discussion to avoid this abrupt ending. Please see our response to reviewer 2 comment.

6) In the Abstract, the authors write 'This new methodological approach can illuminate how miniaturized components interact and function in complex movements of small animals with the potential for 3-D printing of relevant structures.' The potential for 3D printing is not mentioned anywhere else in the manuscript. This topic should be included in the Discussion section of the manuscript.

We have deleted the mention of 3D printing.

Reference

Michels J, Gorb SN (2012) Detailed three-dimensional visualization of resilin in the exoskeleton of arthropods using confocal laser scanning microscopy. Journal of Microscopy 245, 1-16

We have added this reference.

[Editors' note: further revisions were requested prior to acceptance, as described below.]

Reviewer #1:

*This manuscript has greatly improved. I appreciate the level of detail the authors used to incorporate reviewer suggestions. In particular, the Introduction now prepares the readers for work discussed in the following sections. I feel the Discussion section is still a bit long and abruptly ends, however I recommend this manuscript for publication. One small note, in subsection “Distribution of resilin in the energy stored of adult planthoppers”, family names are used, rather than the common names used prior to these sentences. It would be helpful to add the common names here so readers not versed in insect taxonomy can follow.*

Our problem is that there are no common names for these two families or their members. A compromise that we have adopted is to call them Derbid planthoppers and Acanaloniid planthoppers which thus includes the family name and the general group name of planthoppers.

Reviewer #2:

The authors clearly improved the manuscript. However, there are still a few inconsistencies in the context of the confocal laser scanning microscopy (CLSM) visualizations. These are described below.

1) In the first version of the manuscript, the authors wrote “For the imaging of Calcofluor white and Acridine Orange- stained cuticle, laser lines of 405 and 488 nm were used, respectively.” Based on this description, I assumed that the exoskeleton parts with large proportions of chitin had in all cases been visualized by using Calcofluor White M2R and that the exoskeleton parts with large proportions of resilin had in all cases been visualized by using Acridine Orange. And I was already wondering for which visualization the authors had used the autofluorescence that was mentioned in the sentence (but mentioned nowhere else). In the current version of the manuscript, the description is different. The authors now write “For the imaging of Acridine Orange- stained cuticle, laser lines of 405 and 488 nm were used. Autofluorescence of chitin was recorded in the 410-465 nm range;.…” And Calcofluor White M2R is now mentioned (2) in the context of the visualization of muscles and exoskeleton parts with large proportions of chitin (subsection “Imaging and rendering) and (1) in the lines where the authors write 'In preparations where the 405 nm laser-induced autofuorescence of exoskeletal elements, such as fine tendons in nymphs, was not strong, a chitin-binding dye Calcofluor White (0.1 mg/ml, Σ-Aldrich, F3543-1G) was included in the staining procedure.' These descriptions make the reader think that the exoskeleton parts with large proportions of chitin were visualized mainly by using autofluorescence and only in the case of too low autofluorescence intensities by using Calcofluor White M2R. Which version is correct? In this context, I have the following questions:

We agree that the first version of our manuscript lacked precision about the imaging procedure of Acridine Orange-stained samples and a different set of samples stained with phalloidin and Calcofluor White. We amended that in the second version, explaining that Acridine Orange and Calcofluor white were never used simultaneously. To make this even clearer we have done three things:

1) We have reformatted Table 1 to make clearer the distinction between the staining methods.

2) We say “In preparations where the 405 nm laser-induced autofuorescence of exoskeletal elements, such as fine tendons in nymphs, was not strong, a chitin-binding dye Calcofluor White (0.1 mg/ml, Σ-Aldrich, F3543-1G) was included in the staining procedure. Acridine orange and Calcofluor White were never used in the same preparation.”

3) We say “Autofluorescence of chitin was recorded in the 410-465 nm range whereas fluorescence of Acridine Orange was recorded in the 520-580 nm range.”

*a) Why did the authors use an excitation wavelength of 405 nm and why did they detect fluorescence light with wavelengths of 410-465 nm to visualize exoskeleton parts with large proportions of chitin? The application of quite similar settings (405 nm and 420-480 nm) had been demonstrated to result in the visualization of exoskeleton parts with large proportions of resilin (see Michels and Gorb 2012) but not in the visualization of exoskeleton parts with large proportions of chitin. (In a comparable way, the authors used excitation wavelengths of 352-402 nm and detected fluorescence light with wavelengths of 417-477 nm for their wide-field fluorescence microscopy visualization of the resilin autofluorescence (Subsection “Sample preparation”).) Accordingly, with these settings, the authors did very likely not visualize exoskeleton parts with large proportions of chitin.*

We did not notice substantial differences in the distribution of blue, green and red autofluorescence in cuticular samples. Similarly, the boundaries of the detected wavelengths did not have a marked effect on visualization of the energy stores.

We now say: “We did not notice substantial differences in the distribution of blue, green and red autofluorescence in cuticular samples.”

*b) In subsection “Imaging and rendering”, the authors write 'Autofluorescence of cuticle was excited with a 561 nm laser line.' This autofluorescence is also mentioned in the Table 1. However, it is mentioned nowhere else, and the authors do not describe for which visualization this autofluorescence was used.*

We added a sentence clarifying that red autofluorescence was used for visualization of chitinous elements of energy stores alone (i.e. when muscles and resilin were not simultaneously imaged).

“Red autoflorescence was preferentially used for visualization of chitinous elements of the energy stores alone (i.e. when muscles and resilin were not simultaneously imaged), as longer wavelengths are less affected by cuticle pigmentation and thickness, yielding a more uniform signal throughout the sample. Autofluorescence of cuticle was excited with a 561 nm laser line and recorded in the 568-683 nm range.”

We also included the information about using red autofluorescence in relevant figure legends.

2) The authors did not really be responsive to my concerns about the large probability of crosstalk having occurred during the CLSM visualizations. They included information about the detected fluorescence wavelengths in the manuscript, and it is now obvious that the applied detection settings had very likely resulted in the occurrence of crosstalk during the CLSM analyses. For example, the Acridine Orange fluorescence, which was used to visualize the exoskeleton structures with large proportions of resilin, was visualized by applying an excitation wavelength of 488 nm and by detecting fluorescence light with wavelengths of 520-580 nm. However, these settings are very appropriate to also visualize an autofluorescence that is typically excited by weakly or non-sclerotised exoskeleton structures with large proportions of chitin (see Michels and Gorb 2012). Accordingly, the authors cannot be sure that they only visualized structures with large proportions of resilin. In the case of the Calcofluor White M2R fluorescence, which was used to visualize the exoskeleton structures with large proportions of chitin, a similar problem exists: this fluorescence was visualized by applying an excitation wavelength of 405 nm and by detecting fluorescence light with wavelengths of 410-570 nm. With these settings, the autofluorescence of resilin was included in the visualization (see above), and the authors cannot be sure that they only visualized structures with large proportions of chitin.

It is now of course not possible to get rid of this 'crosstalk problem' anymore. The fluorescences of fluorescence dyes are often much more intense than autofluorescences. In such situations, the proportions of autofluorescences within the fluorescence signal can be minor, and it can happen that the autofluorescences do not negatively affect the analyses. In my opinion, it seems that this was the case in the present study. The results are convincing, and it seems that crosstalk did not play an important role and have a negative effect. However, the authors should mention crosstalk. They should include a statement saying that with the CLSM settings that they applied crosstalk might have occurred but, because of the relatively high intensities of the dye fluorescences compared to those of the autofluorescences, very likely did not negatively influence the results.

We agree that some crosstalk may occur in visualizing the cuticular elements of the energy stores. We already described our approach to reducing the chitin autofluorescence component during visualization of Acridine Orange-stained resilin. We now added the following text for further clarification:

“Using these wavelengths for visualization of calcofluor white-stained exoskeleton and Acridine Orange (AO) -stained resilin creates the possibility of crosstalk. […]The intensity of AO fluorescence is much stronger than autofluorescence of chitin, however, so that the crosstalk did not play an important role in our analysis of the distribution of resilin and did not impact the results.”

Our conclusion is thus in agreement with that of the reviewer “because of the relatively high intensities of the dye fluorescences compared to those of the autofluorescences, very likely did not negatively influence the results.”

3) The authors should remove the sub-figures Aiii and Biii from the 'Figure 3—figure supplement 1' because of the inconsistencies regarding the correct visualization of the resilin autofluorescence (see above). Based on the appearance of these sub-figures, I cannot imagine that the respective results are correct.

We have removed panels Aiii and Biii from the supplementary figure and all references to them in the text.

4) In the subsection “Imaging and rendering”, the authors should not use the term 'contamination'.

We replaced “contamination” with “contribution”.